# The Sensitivity of Tau Tracers for the Discrimination of Alzheimer’s Disease Patients and Healthy Controls by PET

**DOI:** 10.3390/biom13020290

**Published:** 2023-02-03

**Authors:** Zohreh Mohammadi, Hadi Alizadeh, János Marton, Paul Cumming

**Affiliations:** 1Immunology Research Center, Tabriz University of Medical Sciences, Tabriz 5166/15731, Iran; 2Student Research Committee, Tabriz University of Medical Sciences, Tabriz 5166/15731, Iran; 3ABX Advanced Biochemical Compounds Biomedizinische Forschungsreagenzien GmbH, Heinrich-Glaeser-Straße 10-14, D-01454 Radeberg, Germany; 4Department of Nuclear Medicine, Bern University Hospital, Freiburgstraße 18, CH-3010 Bern, Switzerland; 5School of Psychology and Counselling, Queensland University of Technology, Brisbane, QLD 4059, Australia

**Keywords:** Alzheimer’s disease, tau neurofibrillary tangles (NFTs), positron emission tomography

## Abstract

Hyperphosphorylated tau aggregates, also known as neurofibrillary tangles, are a hallmark neuropathological feature of Alzheimer’s disease (AD). Molecular imaging of tau by positron emission tomography (PET) began with the development of [^18^F]FDDNP, an amyloid β tracer with off-target binding to tau, which obtained regional specificity through the differing distributions of amyloid β and tau in AD brains. A concerted search for more selective and affine tau PET tracers yielded compounds belonging to at least eight structural categories; ^18^F-flortaucipir, known variously as [^18^F]-T807, AV-1451, and Tauvid^®^, emerged as the first tau tracer approved by the American Food and Drug Administration. The various tau tracers differ concerning their selectivity over amyloid β, off-target binding at sites such as monoamine oxidase and neuromelanin, and degree of uptake in white matter. While there have been many reviews of molecular imaging of tau in AD and other conditions, there has been no systematic comparison of the fitness of the various tracers for discriminating between AD patient and healthy control (HC) groups. In this narrative review, we endeavored to compare the binding properties of the various tau tracers in vitro and the effect size (Cohen’s d) for the contrast by PET between AD patients and age-matched HC groups. The available tracers all gave good discrimination, with Cohen’s d generally in the range of two–three in culprit brain regions. Overall, Cohen’s d was higher for AD patient groups with more severe illness. Second-generation tracers, while superior concerning off-target binding, do not have conspicuously higher sensitivity for the discrimination of AD and HC groups. We suppose that available pharmacophores may have converged on a maximal affinity for tau fibrils, which may limit the specific signal imparted in PET studies.

## 1. Introduction

By definition, tauopathy refers to a group of brain disorders in which intracellular deposition of misfolded hyperphosphorylated tau filaments is a predominant histopathological feature [1]. Tauopathies can arise in various contexts, including traumatic brain injury, and a number of spontaneous neurodegenerative disorders. The ratio of aggregated tau isoforms defines three main groups of tauopathies: 4R (including progressive supranuclear palsy (PSP), corticobasal degeneration (CBD), and argyrophilic grain disease), 3R (Pick’s disease), and 3R+4R tauopathies (AD, Down’s syndrome, chronic traumatic encephalopathy) [2,3,4]. In theory, isoform-selective tau ligands for molecular brain imaging by positron emission tomography (PET) might differentiate between these tauopathies. In practice, the regioselectivity of tau-PET findings serves to identify the presence of AD-typical tau deposition. In this narrative review, we focus on the application of tau-PET for identification of pathology in patients with Alzheimer’s dementia (AD), with the particular objective of summarizing the fitness and sensitivity of available tau tracers for the detection of AD pathology.

The accumulations of tau and Aβ in and around cortical neurons are key features of AD pathology [5]. Tau deposits in AD consist of NFTs, neuropil threads, and neurites in senile plaques, which accumulate in neuronal bodies, dendrites, and axons, respectively [6]. NFTs are ultrastructurally composed of paired helical filaments (PHFs) and lesser amounts of straight filaments (SFs) consisting of all six isoforms of tau protein [7,8,9]. Although the immunoreactivities of tau aggregates are similar in various tauopathies, the isoform types have histopathological and ultrastructural differences [10]. Progression of cortical tau pathology in AD brains occurs following Braak staging, traditionally assessed by post mortem examination [11]. In Braak stages I and II, tauopathy is confined to the transentorhinal area and hippocampus, extending in stages III and IV to involve the frontal, parietal, and temporal cortex, and in stages V and VI to the entire neocortex, including motor and sensory core fields. Aggregation of NFTs and consequent perturbation of axonal microtubule structure may precede the initiation of cognitive decline in AD patients [12,13]. Indeed, while early Aβ deposition can accompany normal senescence [14], NFT aggregation does not occur in cognitively healthy people. NFT deposition, which proceeds according to Braak staging, correlates clearly with the degree of neuropathy in AD patients [15,16,17,18,19], also in contrast to the case for Aβ [15], making tau a rational target for molecular brain imaging of AD.

A new era of PET began with the discovery that the amyloid-β (Aβ) ligand 2-(1-(6-[(2-[^18^F]fluoroethyl) (methyl)amino]-2-naphthyl)ethylidene)malononitrile ([^18^F]FDDNP (Figure 1a)) presented off-target binding to tau aggregates in the brains of patients with AD [20]. The target selectivity of [^18^F]FDDNP PET is imparted by the differing distributions of Aβ and the tau protein aggregates known as neurofibrillary tangles (NFTs) in AD and certain other tauopathies. Early results in AD with the breakthrough tracer [^18^F]FDDNP (Figure 1a) inspired a very broad search for tracers with superior selectivity and sensitivity for diagnosis and monitoring of AD pathology. Subsquent years have seen the development of diverse tau-tracers belonging to various structural classes [21], and there are numerous reviews of the prospects for diagnostic tau PET imaging [22,23,24]. Indeed, [^18^F]-flortaucipir, known variously as [^18^F]-T807, AV-451, and Tauvid^TM^, has obtained FDA approval for diagnostic use [25]. 

Meanwhile, there has been a proliferation of tau PET tracers with roughly similar binding properties and sensitivity for the detection of tau pathologies, with little agreement as to the optimal agent for the specific task of AD diagnosis and progression monitoring. This circumstance resembles the surfeit of molecular imaging tracers for the neuroinflammation marker TSPO, many of which have similar binding properties [26].

Herein, we summarize the binding properties of the various tau PET ligands, proceeding from [^18^F]-FDDNP (Figure 1a) to the tracers of other structural classes that have been tested in AD patients. For this purpose, we compiled the available data about the affinity and specificity of the various ligands for tau aggregates in vitro, placing emphasis on the off-target binding of certain classes of tau tracers. As an objective measure of effect size in PET studies, we report Cohen’s d for the contrast between tracer binding in groups of healthy controls (HCs) and AD patients. We intend thereby to establish a rational basis for selecting among the diverse structural classes of tau tracers available at the time of writing.

### Molecular Description of Tau Aggregates

Tau is a naturally unfolded tubulin-binding phospho-protein that promotes the assembly of microtubules and their stabilization [27]. The preponderance of tau expression in the central nervous system (CNS) is in neuronal axons, glia, extracellular space [28,29], and dendrites [30]. Tau is encoded by the microtubule-associated protein tau (MAPT) gene located on human chromosome 17q21, which primarily contains 16 exons [31,32]. Six isoforms of human tau protein, ranging in length from 352 to 441 amino acids, arise through alternative mRNA splicing [33,34], resulting in forms with three or four tandem repeats, generally designated as 3R and 4R [33,35].

3R and 4R isoforms have approximately similar concentrations in the healthy human brain [36,37], but tauopathies can entail a disturbed balance of 3R and 4R tau expression, which may arise from the dysregulation of exon 10 splicing [38]. In vitro, tau binds to microtubules with an affinity (K_D_) in the range of 75–450 nM [39]. However, the additional repeat domain of 4R tau imparts a considerably higher affinity for microtubules in comparison with 3R tau, such that the 4R form has a more prominent role in microtubule assembly [37,39].

A normal tau protein molecule is naturally unfolded and has a low degree of secondary structure [28,40]. Similar to Aβ aggregates, tau filaments contain cross β-sheet structures, with cores consisting of approximately 90 amino acids [41]. The formation of pathological aggregates of tau protein is linked to hyperphosphorylation; in this condition, tau loses its affinity for microtubules and instead forms aggregates known as NFTs [42,43]. Of some 85 Ser/Thr/Tyr residues in the longest CNS tau isoform, nearly half can be phosphorylated [44,45], such that tau NFTs cannot be considered as a singular molecular entity. This phenomenon has doubtlessly complicated the search for an optimal PET tracer for the detection of tau aggregates.

## 2. Development of Tau Tracers for PET

PET employs tracer molecules usually labelled with the short-lived radionuclides [^11^C] (t_1/2_ = 20.4 min) and [^18^F] (t_1/2_ = 109.8 min). An ideal brain radiotracer for tau should fulfill various criteria [46,47]: 1. ability to cross the blood–brain barrier (BBB) and cell membranes; 2. high affinity and selectivity for the target molecule, with low off-target binding, i.e., in the present context >10-fold selectivity for tau over Aβ and low nonspecific binding; 3. lipophilicity within the narrow range of LogP 2.0–3.5; 4. lack of radiolabeled metabolites that can enter the brain; 5. suitable pharmacokinetics/dynamics for rapid PET imaging. We begin this account with a detailed description of the first tau tracer [^18^F]FDDNP (Figure 1a), which was originally described as an Aβ ligand; this history highlights the difficulty presented by the relatively non-selective interaction of ligands with β-sheets, which are present in various biomolecules.

### 2.1. [^18^F]FDDNP, the Accidental First Tau PET Tracer

Development of the first tau tracer began with a fluorescent dye, 1,1-dicyano-2-[6-(dimethylamino)naphthalen-2-yl]propene (DDNP, Figure 1b), originally synthesized for use with fluorescence spectroscopy. DDNP (Figure 1b) possesses an electron donor part (pi-conjugation of a dialkylamino group) and an electron acceptor part (2-cyano acrylate unit) that together impart inherent fluorescence properties. DNNP (Figure 1b) can visualize plaques and tangles through confocal fluorescence microscopy, which encourages the preparation of its fluorinated version, [^18^F]FDDNP (2-(1-(6-[(2-[^18^F]fluoroethyl) (methyl)amino]-2-naphthyl)ethylidene)malononitrile, Figure 1a) [48]. FDDNP is more lipophilic than DNNP (Figure 1b), which might impart relatively higher nonspecific binding, i.e., in white matter [49,50]. The physical properties of DDNP (Figure 1b) led researchers to investigate its binding of fibrillar, insoluble protein aggregates with β-sheet conformations such as Aβ and tau. The binding of the 6-dialkylamino naphhtalenyl-2-cyanoacrylate scaffold of DDNP (Figure 1b) to tau-derived binding sequences has been investigated using X-ray microcrystallography [51]. The partially apolar binding cavity in the VQIVYK sequence (a hexapeptide motif responsible for tau aggregation) of tau protein has a tube-like shape formed between a pair of β-sheets, which may favor the binding of DNNP (Figure 1b) and other uncharged molecules such as benzodiazepines and anesthetics.

#### 2.1.1. In Vitro Studies

The first binding studies with DDNP/FDDNP were undertaken using confocal fluorescence microscopy of brain specimens in vitro [20,52]. Initial results suggested the presence of two kinetically distinguishable binding sites at Aβ fibrils in vitro, with K_D_s of 0.12 nM and 1.86 nM for FDDNP, and somewhat lesser corresponding affinities for its analog, FENE (Figure 1d, 1-(6-[(2-fluoroethyl)(methyl)amino]naphthalen-2-yl)ethenone). In addition, FENE (Figure 1d) had a high-capacity, low-affinity binding site, which tended to obscure the high-affinity site. These results led to the selection of FDDNP for further research as an Aβ tracer. However, autoradiography with [^18^F]FDDNP (Figure 1a) and [^18^F]FENE (Figure 1e, 1-(6-[(2-[^18^F]fluoroethyl)(methyl)amino]naphthalen-2-yl) ethanone) showed binding to tau in AD cortical specimens.

Due to epidemiological findings that non-steroidal anti-inflammatories (NSAIDs) may reduce the risk of AD, researchers [50] tested the interaction between NSAIDs and [^18^F]FDDNP (Figure 1a) at Aβ fibrils from AD brain specimens. There was some displacement by NSAIDs, but not by Congo-red and thioflavin T, which raised a flag about the specificity of [^18^F]FDDNP (Figure 1a) for NFT versus Aβ. [^18^F]FDDNP (Figure 1a) showed binding to Aβ42 (K_D_ = 5.5 nM and B_max_ = 0.28 pmol/nmol) and also to K18ΔK280 tau fibrils (K_D_ = 37 nM and B_max_ = 2.1 pmol/nmol) [53]. With the B_max_/K_D_ ratios being roughly similar for the two targets, these findings in vitro might predict ambivalence of [^18^F]FDDNP (Figure 1a) binding in living brains. Nonetheless, Harada et al. showed higher bind affinity of [^18^F]FDDNP (Figure 1a) for Aβ fibrils than for tau fibrils, as likewise seen for [^3^H]PiB [53]. In 2006, Smid et al. reported on FDDNP labeling in fixed, paraffin-embedded brain tissue sections from patients dying from AD and various other neurodegenerative diseases [54]. Their results indicated that FDDNP preferentially labels structures with Aβ-like histochemical properties. Since Aβ, tau, α-synuclein, and prion fibrils all contain similar β-sheet structures, FDDNP labeling potentially indicates a broad-spectrum of pathologies in vitro, but with preferential binding to tau in AD brains. Others have reported lower sensitivity of FDDNP for Aβ-containing structures when applied at the tracer concentrations typical of PET applications [55]. In 2009, Thompson et al. compared the binding of [^3^H]FDDNP in brain sections from AD patients to that of the specific Aβ ligands [^3^H]PIB and [^14^C]SB13 [56]. There was only weak [^3^H]FDDNP labeling in neocortical tissue sections that contained extensive amyloid plaques and cerebrovascular amyloid angiopathy, which were readily labeled with the Aβ ligands. The authors attributed this to the low affinity of [^3^H]FDDNP for Aβ in AD brain sections. Another study [57] comparing the binding of [^18^F]FDDNP (Figure 1a) and [^3^H]FDDNP to tau and Aβ in vitro suggested that the extreme metabolic instability of [^3^H]FDDNP may have compromised the study by Thompson et al. [56].

Molecular simulations predicted an interaction between FDDNP and monoamine oxidase B (MAO-B). The moderate binding affinity of FDDNP to MAO-B in silico (K_i_ = 99 nM) closely matched that of safinamide (a known MAO-B inhibitor) to the same target (K_i_ = 86 nM) [58]. It remains unknown if MAO-B binding contributes to the [^18^F]FDDNP (Figure 1a) PET signal in living brains, but, as shall be seen, it is certain that other tau tracers do present significant off-target binding to MAO-A or B.

#### 2.1.2. In Vivo PET Studies of [^18^F]FDDNP in AD

[^18^F]FDDNP (Figure 1a) was the first PET tracer to detect tau pathology in the brain of living AD patients [52], showing increased cortical binding in the lateral and medial temporal lobes in AD patients [20]. Based on the preclinical findings described above, the researchers attributed the PET signal to the composite of NFTs and amyloid plaques. Their PET quantitation relative to the pons reference region (SUVr) showed the highest specific binding in the patients’ hippocampus, amygdala, and entorhinal cortex, namely the regions with NFT deposition in AD [11]. The high binding regions matched with brain regions showing hypometabolism measured with [^18^F]FDG-PET, and brain atrophy measured with MRI. A large-scale study tested the fitness of [^18^F]FDDNP (Figure 1a) versus [^18^F]FDG-PET for discriminating healthy elderly, MCA, and AD groups [59]. Results indicated superior group discrimination with [^18^F]FDDNP (Figure 1a) PET compared with structural MRI or FDG-PET. Among three studies (Table 1), the highest Cohen’s d (3.53) was calculated for lateral temporal cortex in the contrast between healthy controls and AD patients with advanced disease (mean MMSE score = 13), with lesser effect sizes in the studies with less severely afflicted patient groups.

Shin et al. directly compared the regional binding of the Aβ-ligand [^11^C]PIB (Figure 1f) and [^18^F]FDDNP (Figure 1a) in the same AD subjects [62]. They found sparse [^11^C]PIB (Figure 1f) binding, but abundant [^18^F]FDDNP (Figure 1a) binding, in the medial temporal cortex, whereas both tracers had high uptake in many neocortical areas, consistent with the overlapping distributions of the two markers in AD. Cognitively normal elderly subjects in that study also had significant [^18^F]FDDNP (Figure 1a) binding in the medial temporal cortex, which correlated inversely with memory scores in the California Verbal Learning Test. Another dual tracer PET study [61] tested for correlations between [^11^C]PIB (Figure 1f) and [^18^F]FDDNP (Figure 1a) binding and CSF markers, finding that [^18^F]FDDNP (Figure 1a) binding correlated directly with CSF levels of Aβ and tau, whereas global [^11^C]PIB signal (Figure 1f) correlated inversely with the CSF Aβ level. Thus, [^18^F]FDDNP (Figure 1a) PET had some degree of specificity toward intracerebral tau in AD. Others called for caution in the interpretation of [^18^F]FDDNP (Figure 1a) results due to its low specific binding to tau [96]. With this caveat, Tolboom et al. reported higher [^18^F]FDDNP (Figure 1a) uptake in AD patients than in HCs, but no difference in their MCI group (n = 11); global cortical [^18^F]FDDNP (Figure 1a) uptake in AD patients had considerable overlap with the range of HC values, which they attributed to the high off-target binding. In human subjects without dementia, impaired cognition, old age, and APOE-4 carriage were associated with higher binding of [^18^F]FDDNP (Figure 1a) in medial and lateral temporal lobes and frontal cortex regions [97]. The partial selectivity of [^18^F]FDDNP (Figure 1a) for NFTs rather than Aβ plaques in AD brains was further suggested in a dual tracer [^18^F]FDDNP (Figure 1a) and [^11^C]PIB (Figure 1f) PET study comparing frontotemporal dementia and AD patients [98], which suggested that [^18^F]FDDNP (Figure 1a) retention was more indicative of tauopathy in AD patients. Due to its imperfect target selectivity, [^18^F]FDDNP (Figure 1a) has largely given way to new PET tracers.

### 2.2. Arylquinoline and Arylquinoxaline Derivatives

In 2005, arylquinoline derivative tracers 2-[(4-[^11^C]methylamino)phenyl]quinoline ([^11^C]BF-158, *N*-[^11^C]methyl-4-(quinolin-2-yl)aniline, Figure 2e)) and 2-(4-aminophenyl)quinoline (BF-170 (Figure 2b)) were discovered by Okamura et al. [99] through the screening of a chemical compound library of small molecular weight β-sheet binding compounds. Although Okamura et al. did not confirm their fitness for human PET studies, their compounds had excellent brain uptake and rapid clearance in living mice. BF-158 (Figure 2a, *N*-methyl-4-(quinolin-2-yl)aniline) and BF-170 (Figure 2b, 2-(4-aminophenyl)quinoline) had somewhat lower EC_50_ values for tau fibrils (399 and 221 nM) compared with Aβ (659 and 786 nM). Hence, BF-158 (Figure 2a) and BF-170 (Figure 2b) displayed higher binding affinity to tau filaments and lower binding affinity to Aβ fibrils than BF-168 (a styrylbenzoxazole derivative, Figure 2c, (*E*)-4-(2-(6-(2-fluoroethoxy)benzo[d]oxazol-2-yl)vinyl)-*N*-methylaniline). The ligands readily visualized NFTs, neuropil threads, and PHF-type neuritis in brain sections, and [^11^C]BF-158 (Figure 2e) autoradiographically labeled NFTs in AD brain sections. However, in a fluorescence assay, these ligands also labeled Aβ fibrils, thus indicating incomplete specificity toward tau pathology. Furthermore, the ligands did not bind significantly to PHFs in non-AD pathologies, suggesting that they might serve for the differentiation of AD from non-AD tauopathies by PET.

#### 2.2.1. [^18^F]THK-523 (BF-242)

Based on preceding results, Fodero-Tavoletti et al. [100] developed [^18^F]THK-523 (Figure 2f, 2-(4-aminophenyl)-6-(2-[^18^F]fluoroethoxy)quinolone; [^18^F]BF-242), the first [^18^F]-labeled arylquinoline tau tracer, from a structural modification of BF170 (Figure 2b). The addition of the alkyl ether group to position C6 of the arylquinoline rendered the structure more selective to tau aggregates and enabled the incorporation of [^18^F]. [^18^F]THK-523 (Figure 2f) along with its congeners [^18^F]THK-5105 (Figure 3b, 1-((2-(4-dimethylamino)phenyl) quinoline-6-yl)oxy)-3-[^18^F]fluoropropan-2-ol) and [^18^F]THK-5117 (Figure 3d, 1-[^18^F]fluoro-3-[2-[4-(methylamino)phenyl]quinolin-6-yl]oxypropan-2-ol) were chosen to proceed into clinical studies. [^18^F]THK-523 (Figure 2f) is of low molecular weight (282.3 g/mol) and has a favorable logP value (2.9 ± 0.1 nM), consistent with high brain uptake.

Initial in vitro saturation studies exhibited two classes of binding sites on synthetic heparin-induced tau polymers (HITP) and one class on Aβ_1-42_ fibrils. [^18^F]THK-523 (Figure 2f) bound biphasically to synthetic HITP with affinities of 1.7 and 22 nM, but to a single site for Aβ_1-42_ with 21 nM affinity. Autoradiographic analysis of human hippocampal serial sections of AD brains demonstrated overlap between [^18^F]THK-523 (Figure 2f) binding and tau immunostaining pattern.

Harada et al. [53] confirmed the higher affinity of [^18^F]THK-523 (Figure 2f) for tau fibrils than for Aβ fibrils. In vitro autoradiography of brain sections from AD patients exhibited a band-like distribution of [^18^F]THK-523 (Figure 2f) in the inner layer of the temporal cortex, also matching the distribution of tau, but not Aβ-immunostaining or [^11^C]PiB (Figure 1f) autoradiography. The frontal gray matter with abundant Aβ plaques had considerably lower [^18^F]THK-523 (Figure 2f) binding compared with [^11^C]PiB (Figure 1f) and [^11^C]BF-227 (Figure 2g, (*E*)-5-(2-(6-(2-fluoroethoxy)benzo[d]oxazol-2-yl)vinyl)-*N*-methyl-*N*-[^11^C]methyl-thiazol-2-amine), whereas the converse was true for the NFT-rich hippocampal CA1 area in the AD brain [101]. [^18^F]THK-523 (Figure 2f) binding in the entorhinal cortex corresponded better to Gallyas silver staining than to tau immunostaining, indicating better labeling of extracellular tau (ghost tangles) [47]. The abundant binding of [^18^F]THK-523 (Figure 2f) to the CA1 area of the AD hippocampus was also confirmed by Tago et al., who reported a pattern matching the numerous tau immune-positive NFTs [102]. In another study, [^3^H]THK-523 bound to AD brain homogenates with an apparent K_D_ of 3.5 nM [103]. Molecular docking studies indicated a lower affinity of THK-523 (Figure 2d) to MAO-B compared with other quinoline derivatives [104].

Another in vitro competition assay revealed low affinity of [^18^F]THK-523 (Figure 2f) to recombinant tau fibrils (K_i_ = 59 nM) and PHF (K_i_ = 87 nM) in AD brain homogenates. This lead the researchers to consider [^18^F]THK-523 (Figure 2f) as a nonselective tau tracer and to note that synthetic tau filaments are an imperfect binding model of native tau filaments in vivo [105]. Immunohistochemical and THK-523 (Figure 2d) fluorescence studies of AD brain sections by Fodero-Tavoletti et al. showed selective binding to argyrophilic tau filaments (PHF-tau) in the hippocampus and frontal regions, but little binding for non-AD tauopathies [106]. This might reflect the differing types of ultrastructure in tau fibrils [36]. Another study showed faint labeling of Aβ plaques by [^18^F]THK-523 (Figure 2f) in AD brain sections, but no labeling of α-synuclein deposits in Lewy bodies of substantia nigra post mortem samples from patients with Parkinson’s disease [106]. Although incomplete specificity might limit the clinical utility of [^18^F]THK-523 (Figure 2f) PET, its selectivity for one ultrastructural form of tau (PHFs forming NFTs) might allow useful differentiation between disease entities.

#### 2.2.2. PET Studies with [^18^F]THK-523

MicroPET analysis demonstrated a significant correlation between the cerebral retention of [^18^F]-THK-523 (Figure 2f) in rTg4510 tau transgenic mice relative to wildtype and Aβ (APP/PS1) transgenic mice and the subsequent tau immunostaining [100,106]. A mouse PET study by Villemagne et al. [47] demonstrated high brain uptake and rapid clearance, along with the absence of lipophilic metabolites. The first-in-man PET study [47] showed greater white matter retention of [^18^F]THK-523 (Figure 2f) compared with gray matter in HC participants, but elevated gray matter binding in widespread neocortical regions and hippocampus of AD patients. The binding pattern in patients correlated with PHF-tau distribution in the post mortem brain, being higher in temporal and parietal lobes than in frontal regions. The retention of [^18^F]THK-523 (Figure 2f) in the hippocampus and insula of HCs without dementia with high Aβ accumulation to [^11^C]PiB (Figure 1f) PET resembled that of AD patients, but cortical retention of [^18^F]THK-523 (Figure 2f) was otherwise higher in AD patients. In the whole neocortex, which represents the spatial distribution of tau in Braak V/VI, the tracer had a Cohen’s d of 3.42 for discriminating AD patients and HCs (Table 1). Hippocampal retention correlated inversely with hippocampal atrophy and several cognitive parameters. However, white matter retention of [^18^F]THK-523 (Figure 2f) interfered significantly with the detection of cortical tau in AD patients, which might indicate native tau or PHF deposits in the white matter of AD subjects [107]. The resultant need for partial volume correction of tau PET images was discouraging for further clinical use of the tracer [47].

#### 2.2.3. [^18^F]THK-951

A search for compounds with lesser hydrophobicity yielded [^11^C]-THK-951 (Figure 3a, 2-(4-([^11^C]methyl)-amino)phenyl)quinoline-7-ol) through hydroxylation of BF-158 (Figure 2a) [102], which was sufficiently lipophilic to penetrate the BBB (logP 1.28) and showed high affinity for displacing [^18^F]THK-523 (Figure 2f) from AD brain homogenates (K_i_ = 21 nM). Fluorescence binding assays with THK-951 (Figure 3a) showed binding to recombinant K18Δ280K-tau fibrils. AD hippocampus sections showed distinct THK-951 (Figure 3a) fluorescence staining of NFTs and neuropil threads. In an ex vivo biodistribution study in normal mice, [^11^C]THK-951 (Figure 3a) showed very rapid washout from the brain, greatly exceeding that for [^18^F]THK-523 (Figure 2f) (1.9) and [^18^F]THK-5117 (Figure 3d) [23]. Labeling of THK-523 (Figure 2d) with ^11^C instead of ^18^F slightly decreased the binding affinity to tau aggregates, which was compensated by improved brain kinetics, i.e., faster washout. Autoradiography analysis of AD hippocampal sections with [^11^C]THK-951 (Figure 3a) showed high binding in the CA1 region of the hippocampus, in a pattern matching with [^18^F]THK-523 (Figure 2f) binding and tau immunohistochemistry (IHC), but distinct from that of Aβ. [^11^C]THK-951 (Figure 3a) gave a good tradeoff between high binding affinity for tau and appropriate BBB permeability. No further studies have evaluated [^11^C]THK-951 (Figure 3a) and its derivatives due to its relatively low affinity to tau compared with [^18^F]THK-523 (Figure 2f), [^18^F]THK-5105 (Figure 3b), and [^18^F]THK-5117 (Figure 3d).

#### 2.2.4. [^18^F]THK-5105 and [^18^F]THK-5117

Due to the inadequacies of [^18^F]THK-523 (Figure 2f) and [^18^F]THK-951, researchers developed three new 2-arylquinoline derivatives through several optimization processes, namely, [^18^F]THK-5105 (6-[(3-^18^F-fluoro- 2-hydroxy)propoxy]-2-(4-dimethyl aminophenyl) quinoline, Figure 3b), [^18^F]THK-5116 (6-[(3-^18^F-fluoro-2-hydroxy)propoxy]-2-(4-aminophenyl)quinoline, Figure 3c), and [^18^F]THK-5117 (6-[(3-^18^F-fluoro-2-hydroxy) propoxy]-2-(4-methylaminophenyl) quinoline, Figure 3d) [105]. These compounds were derived from BF170 through the addition of a secondary alcohol to the fluoroethoxy chain along with N-dimethylation to yield [^18^F]THK-5105 (Figure 3b) and *N*-monomethylation for [^18^F]THK-5117 (Figure 3d). These modifications improved the pharmacokinetic profiles of the tracers. However, [^18^F]THK-5116 (Figure 3c) was not further studied due to its brain kinetics and lower specific signal in AD brain sections [63].

The logP values were 3.0 for [^18^F]THK-5105 (Figure 3b) and 2.3 for [^18^F]THK-5117 (Figure 3d) [108]. The greater lipophilicity of [^18^F]THK-5105 (Figure 3b) imparted higher nonspecific binding to white matter, brainstem, and thalamus [109,110]. [^18^F]THK-5105 (Figure 3b) showed two classes of binding sites on K18Δ280 K-tau fibrils (K_D1_ = 1.5 nM, K_D2_ = 7.4 nM, B_max1_ = 6.9 pmol/nmol fibrils, and B_max2_ = 20 pmol/nmol fibrils) and one [^18^F]THK-5105 (Figure 3b) binding site for Aβ_1-42_ (K_D1_ = 36 nM) [105]. Medial temporal AD brain homogenates containing a high density of tau and a moderate density of Aβ showed higher affinity of [^18^F]THK-5105 (Figure 3b) (K_D_ = 2.6 nM) and [^18^F]THK-5117 (Figure 3d) (K_D_ = 5 nM) than for [^18^F]THK-523 (Figure 2f) (K_D_ = 87 nM). In vitro autoradiography of AD brain sections with [^18^F]THK-5105 (Figure 3b) and [^18^F]THK-5117 (Figure 3d) at tracer concentrations was also consistent with the tau staining pattern (Gallyas–Braak staining). IHC results matched the [^18^F]THK-5105 (Figure 3b) and [^18^F]THK-5117 (Figure 3d) binding in the deep layers of gray matter in the medial temporal cortex and CA1 area of the hippocampus, but there was a mismatch with the [^11^C]PiB (Figure 1f) binding pattern and Aβ IHC staining. [^18^F]THK-5105 (Figure 3b) and [^18^F]THK-5117 (Figure 3d) did not label the hippocampus of healthy control subjects. [^18^F]THK-5116 (Figure 3c) was not pursued as a tau PET tracer due to its inadequate affinity and high nonspecific binding in AD brain sections. Biodistribution studies of mice suggested that [^18^F]THK-5105 (Figure 3b) and [^18^F]THK-5117 (Figure 3d) had better brain uptake and washout when compared with [^18^F]THK-523 (Figure 2f).

[^3^H]THK-5117 did not bind to MAO-B in vitro in post mortem brain samples from AD patients [110]. However, high nonspecific white matter accumulation was observed for [^18^F]THK-5117 (Figure 3d), likely due to binding to β-sheet structures in myelin [64]. Due to the higher selectivity and specificity of the *S* enantiomer compared with the *R* enantiomer of [^18^F]THK-5117 (Figure 3d) to tau aggregates in AD brains, a new tracer called [^18^F]THK-5317 (Figure 3f, 6-((3-[^18^F]fluoro-2-hydroxy)propoxy)-2-(4-methylaminophenyl) quinoline) was developed, which proved to have superior pharmacokinetics in rodents. However, both enantiomers had a similar distribution in autoradiography studies of post mortem AD brains [108,109].

##### Human PET Studies of [^18^F]THK-5105 and [^18^F]THK-5117

A PET study showed greater gray matter retention and lower white matter retention for [^18^F]THK-5105 (Figure 3b) when compared with [^18^F]THK-523 (Figure 2f) [63]. [^18^F]THK-5105 (Figure 3b) possessed higher peak brain uptake than [^18^F]THK523 (Figure 2f), [^18^F]T807 (see Section 2.4), [^18^F]T808 (see Section 2.4), 2-[4-(2-[^18^F]fluoroethyl)-1-piperidinyl]pyrimido [1,2-a]benzimidazole), and [^11^C]PBB3 (see Section 2.5.1), 2-[(1*E*,3*E*)-4-[6-[^11^C]methylamino)pyridin-3-yl] buta-1,3-dienyl]-1,3-benzothiazol-6-ol), obtaining an SUC of 4.5 in the cerebellum. The tracer showed high non-specific accumulation in subcortical white matter, thalamus, and brainstem. However, tracer retention in subcortical white matter was lower than that of [^18^F]THK523 [12]. In addition, [^18^F]THK-5105 (Figure 3b) had high gray matter background signals resulting from its slow kinetics and high lipophilicity.

The first [^18^F]THK-5105 (Figure 3b) PET study showed differentiation of AD patients from HCs, and mismatch with the binding pattern of [^11^C]PiB (Figure 1f) [63], despite fairly high nonspecific binding of the tau tracer in white matter, brainstem, and thalamus [111]. [^18^F]THK-5105 (Figure 3b) was not investigated further as a tau PET tracer in humans. Being more hydrophilic, [^18^F]THK-5117 (Figure 3d) had faster brain kinetics in mice than did [^18^F]THK-5105 (Figure 3b) [112]. The first [^18^F]THK-5117 (Figure 3d) PET study in AD patients showed higher retention in the temporal lobe, correlating to an extent with the individual degree of cognitive impairment, and not overlapping with [^11^C]PIB (Figure 1f) binding [64]. A longitudinal [^18^F]THK-5117 (Figure 3d) PET study revealed annual increases in tracer retention in the middle and inferior temporal gyri and fusiform gyri of AD patients, which correlated with their declining cognitive function [113]. Compartmental analysis of [^18^F]THK-5317 (Figure 3f) uptake relative to a metabolite-corrected input function indicated reversible binding kinetics in MCI and AD patients [114].

Cross-sectional [^18^F]THK-5317 (Figure 3f) PET studies successfully differentiated MCI and AD patients from HCs, showing a pattern of tracer uptake in accord with post mortem tau studies [115]. Test–retest variability was low (<4%) and white matter uptake was lower than for [^18^F]THK-5117 (Figure 3d). However, off-target binding in the basal ganglia was evident. The high white matter binding of [^18^F]THK-5117 (Figure 3d) and [^18^F]THK-5317 (Figure 3f), probably reflecting β-sheet structures in myelin basic protein, reduces the clinical utility of these tracers [114,116]. The estimated Cohen’s d between AD and HC groups was 3.58 in the inferior temporal cortex for [^18^F]THK-5105 (Figure 3b) [63] (Table 1). Cohen’s d for the contrast between AD and HC groups for [^18^F]THK-5117 (Figure 3d) was as high as 2.74 in the inferior temporal lobe (Table 1) [64].

#### 2.2.5. [^18^F]THK-5351

The high white matter binding of [^18^F]THK-5117 (Figure 3d) and [^18^F]THK-5317 (Figure 3f) (*S* enantiomer of [^18^F]THK-5117 (Figure 3d)) motivated the development of [^18^F]THK-5351 (Figure 3g, GE-216, (2*R*)-1-[^18^F]fluoro-3-[2-[6-(methylamino)pyridin-3-yl]quinolin-6-yl] oxypropan-2-ol), the 4-methyl-aminopyridyl-analog of THK-5117 [116]. Lesser lipophilicity of the new tracer (logP = 2.3) results from the substitution of pyridine at 2-aryl group for benzene, imparts lower nonspecific binding to white matter along with a higher affinity for AD tau compared with [^18^F]THK-5117 (Figure 3d), and higher selectivity for tau over Aβ. [^18^F]THK-5351 (Figure 3g) displayed higher binding affinity (K_D_ = 2.9 nM) to hippocampal homogenates of AD brains than did [^18^F]THK-5117 (Figure 3d), and autoradiography indicated a greater gray vs. white matter binding ratio compared with previous quinoline derivatives. Autoradiography in midbrain sections from the cognitively normal individuals showed saturable binding of [^18^F]THK-5351 (Figure 3g) in the substantia nigra, and a saturation binding assay in melanin-containing cells showed high capacity low-affinity binding sites for the tracer on B16F10 cells (K_D_ = 440 nM) [117]. [^18^F]THK-5351 (Figure 3g) had a binding ratio of 2.5 in the substantia nigra, and competitive binding assays of THK-5351 for [^18^F]THK-5351 (Figure 3g) binding to B16F10 melanin-producing cells revealed moderate affinity (K_i_ = 0.22 µM).

##### Human PET Studies of [^18^F]THK-5351

PET studies have shown a higher gray-to-white ratio and faster kinetics for [^18^F]THK-5351 (Figure 3g) compared with other quinoline derivatives [118]. [^18^F]THK-5351 (Figure 3g) PET in AD patients indicated lower white matter binding and higher binding in the temporal lobe and other regions susceptible to AD neuropathology compared with [^18^F]THK-5117 (Figure 3d) and [^18^F]THK-5317 (Figure 3f) [119,120,121]. Cortical uptake in AD patients did not match the expected NFT pathology pattern. Comparative PET revealed better agreement of the [^11^C]THK-5351 (Figure 3h) distribution in AD patients with known NFT neuropathology compared with [^11^C]PBB3 (Figure 6a) [122].

A 2020 PET study showed off-target binding of [^18^F]THK-5351 (Figure 3g) in the basal ganglia and, to a lesser extent, in the cerebral cortex of healthy volunteers, which was attributed to MAO-B, apparently accounting for the imperfect match with tau pathology [123]. Another study comparing [^18^F]THK-5351 (Figure 3g) with [^18^F]AV-1451 (Figure 5a, 7-(6-[^18^F]fluoro-pyridin-3-yl)-5*H*-pyrido[4,3-b]indole) found more prominent cortical uptake of [^18^F]THK-5351 (Figure 3g) in FTD patients, and higher [^18^F]AV-1451 (Figure 5a) uptake in AD [124]. [^18^F]THK-5351 (Figure 3g) proved less sensitive and specific than [^18^F]AV-1451 (Figure 5a) to tau pathology in AD but also served for the detection of non-AD tauopathies. There was a strong correlation between [^18^F]THK-5351 (Figure 3g) and [^18^F]THK-5317 (Figure 3f) binding patterns in the same individuals [121].

[^18^F]THK-5351 (Figure 3g) did not exhibit significant off-target binding in venous sinuses, basal ganglia, or choroid plexus in some studies [119,120,121], but others reported off-target binding in the basal ganglia, midbrain, and thalamus, and higher affinity to MAO-B compared with [^18^F]THK-5117 (Figure 3d). Its binding correlated with levels of the astrocyte marker GFAP, confirming an association with MAO-B [112]. This off-target binding of [^18^F]THK-5351 (Figure 3g) exceeded that of [^18^F]THK-5317 (Figure 3f) [125]. Indeed, the in vivo distribution of [^18^F]THK-5351 (Figure 3g) correlated with the post mortem distribution of MAO-B. The off-target binding of [^18^F]THK-5351 (Figure 3g) in basal ganglia, white matter, midbrain, and thalamus also exceeded that of [^18^F]AV-1451 (Figure 5a) [124]. Administration of an MAO-B inhibitor reduced the off-target binding of [^18^F]THK-5351 (Figure 3g) by about 50% [126] according to SUV. However, the SUVr index indicated no statistically significant reduction in [^18^F]THK-5351 (Figure 3g) off-target binding [127]. Similar binding patterns between [^18^F]THK-5351 (Figure 3g) and other quinoline derivatives imply a more general interaction of these tracers with MAO-B [125]. A post mortem study of an AD patient who had undergone ante mortem [^18^F]THK-5351 (Figure 3g) PET demonstrated [^3^H]THK-5351 binding to MAO-B, which was compatible with previous PET findings [128], thus arguably disqualifying [^18^F]THK-5351 (Figure 3g) as a selective tau biomarker in AD diagnosis. However, Cohen’s d values for the contrast between AD patients and HCs were as high as 2.44 in the frontal lobe, indicating that the tracer could serve operationally for the detection of AD pathology (Table 1) [129]. Another study [67] reported a Cohen’s d of 3.00 in inferior temporal gyrus for the contrast of AD and HC groups (Table 1).

#### 2.2.6. 2-Phenylquinoxaline Derivatives

[^18^F]-S16 ((*S*)-1-(4-(6-(dimethylamino)quinoxalin-2-yl)phenoxy)-3-[^18^F]fluoro propan-2-ol,^18^F-S16, Figure 3i) is a novel 2-phenylquinoxaline tau tracer [65,95]. Initial thinking was that the 2-phenylquinoxaline scaffold detected β-sheet structures in Aβ, but later in vitro investigations showed binding to NFTs [130]. The optimization of THK tracers by adding a fluoropropanol side chain gave rise to [^18^F]THK-5117 (Figure 3d) and [^18^F]THK-5351 (Figure 3g) [64,116]. The introduction of a chiral 2-fluoromethyl-1,2-ethylenediol side chain to the position-4′ and a -N(CH_3_)_2_ group in position-6 of the 2-phenylquinoxaline scaffold resulted in higher selectivity to tau over Aβ and improved brain kinetics of [^18^F]-S16 (Figure 3i). The 4′-*O*-[(*S*)-1-[^18^F]fluoropropan-2-ol] substituent of [^18^F]-S16 (Figure 3i) increased the hydrophilicity (logD = 2.61, clogP = 3.01) [130]. Radiosynthesis of [^18^F]-S16 (Figure 3i) was achieved in a high-yield, fully automated process [95].

The hydrophilic characteristics of the tracer resulted in the increased selectivity for tau over Aβ (35-fold; K_i_ = 356 nM in competition with [^3^H]PiB) [95]. [^3^H]-S16 had K_i_ = 10 nM in quantitative binding assays of AD brain homogenates in competition with [^3^H]THK-523 binding [130]. However, the tracer only weakly displaced [^3^H]T807, with K_i_ = 611 nM. The IC_50_ of the tracer for MAO-B was >10 μM, as compared with 0.5 μM for THK-5351. Autoradiography of entorhinal cortex from human AD brain sections showed high binding of [^18^F]-S16 (Figure 3i), which was associated with tau fluorescent staining and was displaceable by THK-5371.

MicroPET studies with [^18^F]-S16 (Figure 3i) in rTg4510 mice showed two-fold higher BBB penetration than for [^18^F]THK-5351 (Figure 3g), with notably higher initial brain uptake (11% ID/g) and rapid washout [130]. The tracer did not show in vivo defluorination. Initial human PET studies of [^18^F]-S16 (Figure 3i) in AD showed rapid brain uptake, with slower clearance compared with [^18^F]THK5317 [65,94]. There was substantial hepatobiliary excretion and an absence of defluorination in vivo [95,130]. The parietal lobe, posterior cingulate, and precuneus showed the highest SUVrs in AD patients. Another PET study [94] showed elevated uptake in AD patients compared with HCs, particularly in the bilateral occipital cortex, posterior cingulate cortex/precuneus, and lateral frontal cortex. Although there were no statistically significant differences in SUVr values between AD patients and HCs in the medial temporal lobe [95], there was an overall inverse correlation between [^18^F]-S16 (Figure 3i) binding and brain metabolism to [^18^F]FDG PET. Furthermore, [^18^F]-S16 (Figure 3i) PET differentiated AD and HC groups with Cohen’s d values ranging from 0.63 to 1.78 (Table 1), and an inverse relationship between cortical [^18^F]-S16 (Figure 3i) uptake with MMSE scores [94,95]. The tracer had little binding in white matter, choroid plexus, and the sagittal sinus [94,95]. However, there was some off-target binding of [^18^F]-S16 (Figure 3i) in the lentiform nucleus and thalamus [95]. There was [^18^F]-S16 (Figure 3i) retention in the brainstem, substantia nigra, and basal ganglia of both healthy controls and AD patients, again consistent with off-target binding.

### 2.3. Derivatives of Lanzoprasole and Astemizole

Lansoprazole (Figure 4a, (2-[[3-methyl-4-(2,2,2-trifluoroethoxy)pyridin- 2-yl]methyl sulfinyl]-1*H*-benzimidazole) and astemizole (Figure 4b, 1-[(4-fluorophenyl)methyl]-*N*-[1-[2-(4-methoxyphenyl)ethyl]piperidin-4-yl]benzimidazol-2-amine) are benzimidazole-derivative proton pump inhibitors. They proved to have high affinity for NFTs; the K_i_s for heparin-induced tau filaments were 2.5 nM for lansoprazole (Figure 4a) and 1.9 nM for astemizole (Figure 4b), and corresponding K_i_s to PHF-tau were 2.1 nM and 830 nM, respectively [131]. Other in vitro autoradiography studies of human AD brain sections with *N*-methyl lansoprazole (Figure 4c, NML) labeled with [^11^C] (Figure 4d, [^11^C]NML) and [^18^F] (Figure 4e, [^18^F]NML) revealed a high affinity for tau aggregates in AD, with 12-fold selectivity for tau over A_β_ (K_D_ amyloid/K_D_ tau) [132]. [^18^F]NML (Figure 4e) had a K_D_ of 0.7 nM toward HITF [133,134], whereas autoradiography studies of AD brain homogenates demonstrated binding to 3R and 4R tau with a K_D_ of 8.2 nM [132].

Preclinical in vivo studies of [^18^F]NML (Figure 4e) and [^11^C]NML (Figure 4d) revealed adequate BBB clearance in rhesus macaques, but no entry into the brains of mice unless pretreated with cyclosporine, a P-glycoprotein 1 transporter inhibitor [132,135,136,137]. However, astemizole derivatives were impermeable to the mouse BBB despite cyclosporine treatment [133,136]. An [^18^F]NML (Figure 4e) PET study in healthy humans showed rapid entry and washout in the brain, with a maximum SUV of 3–4 in the gray matter of the cerebellum and cortex [133,134]. There was low tracer retention in the brains of AD, MCI, and PSP patients, such that [^18^F]NML (Figure 4e) was abandoned, despite promising in vitro binding properties [134]. The structure of astemizole-based radiotracers is depicted in Figure 4f.

### 2.4. Benzimidazopyridine and Benzimidazopyrimidine Derivatives: Flortaucipir and T808

[^18^F]T807 (Figure 5a, 7-(6-[^18^F]fluoro-pyridin-3-yl)-5*H*-pyrido[4,3-b]indole), also known as [^18^F]flortaucipir and [^18^F]AV-1451 (Figure 5a), is the first FDA-approved tau tracer, marketed since 2020 under the name of TAUVID^TM^ [25]. It is the most widely studied and used tau tracer, having attracted the attention of many researchers since its first discovery in 2013. [^18^F]flortaucipir (Figure 5a) and its congener [^18^F]T808 (Figure 5b), 2-(4-(2-[^18^F]fluoroethyl)piperidin-1-yl)benzo[[4,5]]imidazo[1,2-a]pyrimidine were discovered through the process of optimization of two other tau ligands, T726 and T557, aiming to increase their selectivity for PHF-tau over Aβ_1-42_ and to improve their pharmacokinetic characteristics [137,138,139].

#### 2.4.1. In Vitro Studies with [^18^F]Flortaucipir and [^18^F]T808

In vitro saturation binding assays of immunopurified PHF-tau from AD brain homogenates indicated a K_D_ of 0.7 nM and B_max_ of 310 pmol/mg [25], whereas autoradiographic studies of the frontal lobe from post mortem AD patients indicated a K_D_ of 15 nM [140] and 14.6 nM [138] for PHF-tau. Combined immunohistochemistry and in vitro autoradiography studies of human AD brain sections revealed that [^18^F]T807 (Figure 5a) and [^18^F]T808 (Figure 5b) both had higher binding to PHF-tau aggregates than to Aβ [138]; [^18^F]Flortaucipir (Figure 5a) had a logP of 1.7, predicting adequate brain penetration and fast washout. Screening studies indicated limited off-target binding to more than 72 CNS targets. In vitro autoradiography revealed colocalization of [^18^F]flortaucipir (Figure 5a) with tau aggregates and no colocalization of the tracer with Aβ, TDP43, and α-synuclein [138,141]. Nonetheless, some studies demonstrated off-target binding of [^18^F]flortaucipir (Figure 5a) to MAO-A and MAO-B [132,142], although this was not reported in other studies [141,143,144]. On the other hand, [^3^H]flortaucipir had a K_D_ of 1.6 nM for human recombinant MAO-A protein [25], as might be expected from the β-carboline motif. Another study reported saturable binding of [^18^F]flortaucipir (Figure 5a) to melanoma cells with K_D_ = 670 nM. The post mortem [^18^F]flortaucipir (Figure 5a) binding pattern in AD brains correlated well with NFT Braak staging [141]. However, specific binding was not identifiable in the hippocampus due to spillover of off-target binding from the nearby choroid plexus [143]. CSF-tau levels correlated with [^18^F]flortaucipir (Figure 5a) retention in brain at early stages of AD, and also in presymptomatic patients, especially in the neocortex and parahippocampal gyrus [142,145,146]. There was off-target binding in the leptomeningeal melanin, substantia nigra, choroid plexus, and cerebral vessels of AD patients [141,143]. [^18^F]flortaucipir (Figure 5a) retention in these regions did not correlate with NFT Braak staging. In vitro binding studies with AD brains also demonstrated off-target binding of the tracer to MAO-A and MAO-B [125,147], although [^18^F]flortaucipir (Figure 5a) exhibited ten-fold lower affinity to MAO-B than did [^3^H]THK-5351. Another PET study in PD patients did not indicate any effect of ongoing treatment with an MAO-B inhibitor [148].

#### 2.4.2. PET Studies with [^18^F]Flortaucipir and [^18^F]T808

BBB penetration of the [^18^F]flortaucipir was confirmed in mouse PET studies showing rapid brain penetration and moderate washout [138]; kidney clearance was the main route of systemic elimination. [^18^F]Flortaucipir (Figure 5a) exhibited minor defluorination in mice, but analysis of brain homogenates did not detect labeled metabolites. Initial PET studies of AD patients with [^18^F]flortaucipir (Figure 5a) [145] and [^18^F]T808 (Figure 5b) [149] revealed their preferential retention in brain areas containing PHF-tau filaments. Besides, both tracers had low white matter binding, thus yielding good contrast between white and gray matter. However, [^18^F]T808 (Figure 5b) PET was not pursued due to its in vivo defluorination in rodents, resulting in cranial labeling [139,140]. [^18^F]Flortaucipir (Figure 5a) retention in AD patients correlated with disease severity, although the tracer uptake showed considerable variability in HCs, especially those younger than 40 years [140]. Another study showed correlations between tracer retention and clinical stage in AD, MCI, and Aβ+ cognitively normal adults [74]. [^18^F]Flortaucipir (Figure 5a) PET differentiated AD patients from MCIs and healthy controls [140,150,151,152,153]. In the contrast between AD and healthy control groups, Cohen’s d ranged from one–two in studies of patients with moderate disease (MMSE circa 20), but was greater than three in a study of advanced disease (Table 1). The tracer retention and distribution pattern in [^18^F]flortaucipir (Figure 5a) PET of AD brains correlated with Braak staging of NFT pathology [154], and the area coverage of [^18^F]AV-1451 (Figure 5a) retention correlated with disease severity [155]. [^18^F]flortaucipir (Figure 5a) PET in conjunction with post mortem pathology reliably detected Braak VI tau pathology, but the tracer was not sensitive to Braak I–IV tau pathology in AD patients compared with young tau-negative HCs [156].

[^18^F]Flortaucipir (Figure 5a) retention occurred mostly in posterior parietal and inferior temporal cortices of AD patients, and there was some correlation between [^18^F]flortaucipir (Figure 5a) retention and impaired cognition, even in seemingly healthy older subjects [145,153]. Early onset AD patients had increased tracer uptake across the neocortex compared with HCs, and late-onset AD patients exhibited significantly elevated tracer uptake mostly in the temporal lobe [157]. [^18^F]Flortaucipir (Figure 5a) PET showed higher retention in Aβ+ patients without dementia, but not in Aβ− cases [74]. Another study revealed higher tracer uptake in early onset AD cases compared with late-onset ones in the premotor and prefrontal cortex and inferior parietal cortex [157].

### 2.5. Pyridinyl-Butadienyl-Benzothiazole (PBB) Derivatives

#### 2.5.1. [^11^C]PBB3

[^11^C]PBB3 (Figure 6a) (2-((1*E*,3*E*)-4-(6-((methyl-^11^C)amino)pyridin-3-yl)buta-1,3-dien-1-yl) benzo[d]thiazol-6-ol) is a first generation tau tracer developed by Higuchi et al. [158] by screening a number of fluorescent compounds binding to β-sheet structures. PBB3 and [^11^C]PBB3 (Figure 6a) showed appropriate brain kinetics and rapid peripheral metabolism in plasma and brain specimens and were selected for further investigation [159]. PBB3 has high selectivity for tau aggregates (K_D_ = 2.6 nM), with a 50-fold higher affinity for tau than for Aβ [159,160]. Initial autoradiography studies showed binding in the CA1 and subiculum regions of the hippocampus of AD patients, and revealed tau pathology in non-AD tauopathies, suggesting a preferred interaction with 4R tau [159,161]. Comparison of [^11^C]PBB3 (Figure 6a) and [^18^F]flortaucipir (Figure 5a) binding in AD brain homogenates revealed distinct selectivity of the former tracer towards various forms of tau fibrils, especially dystrophic neurites and NFTs [162]. There was no off-target binding of [^11^C]PBB3 (Figure 6a) to MAO [160], and the ligand had low affinity for Aβ [163] and α-synuclein aggregates [164]. Nonetheless, [^11^C]PBB3 (Figure 6a) binding seemed to have a greater association with Aβ than was seen with [^18^F]THK5351 (Figure 3g) [122]. Indeed, the distribution pattern of [^11^C]PBB3 (Figure 6a) seemed relatively less indicative of AD tau pathology.

[^11^C]PBB3 (Figure 6a) PET studies in PS19 mice expressing 4R tau isoform mutations revealed rapid brain uptake and tracer binding to NFTs [165]. However, the tracer yielded a lipophilic metabolite in mice, which likely crossed the BBB and contributed to the background signal in vivo [165]. The first [^11^C]PBB3 (Figure 6a) PET study in AD patients demonstrated labeling of medial and lateral temporal cortices and frontal cortex, and good discrimination of AD and HC groups, as well as mismatches with [^11^C]PIB (Figure 1f) binding in the same subjects [159], as was confirmed in another [^11^C]PBB3 (Figure 6a) PET study [166]. The binding of [^11^C]PBB3 (Figure 6a) correlated with cognitive decline and gray matter atrophy in AD patients [77]. [^11^C]PBB3 (Figure 6a) PET gave a Cohen’s d for discrimination of AD patients and HCs in the range of two–three (Table 1).

Off-target binding in the choroid plexus, longitudinal sinus, and basal ganglia, and metabolic instability have limited the human application of [^11^C]PBB3 (Figure 6a) [4]. Furthermore, the brief half-life of carbon-11 disfavors its routine use. [^11^C]PBB3 (Figure 6a) undergoes photoisomerization under fluorescent light, which results in a rapid decrease in purity [167]. In addition, rapid metabolism of [^11^C]PBB3 (Figure 6a) in vivo hinders its brain uptake, while lipophilic radiometabolites may confound the brain signal. Hence, [^18^F]PM-PBB3 (Figure 6c) was developed to overcome these shortcomings.

#### 2.5.2. [^18^F]PM-PBB3

[^18^F]PM-PBB3 (Figure 6c, APN-1607, 1-[^18^F]fluoro-3-((2-((1E,3E)-4-(6-(methylamino) pyridin-3-yl)buta-1,3-dien-1-yl)benzo[d]thiazol-6-yl)oxy)propan-2-ol), also known as [^18^F]florzolotau, is a propylated analogue of [^11^C]PBB3 (Figure 6a) [166,168]. It was proposed to circumvent the off-target binding of [^11^C]PBB3 (Figure 6a), metabolic instability, and the brief half-life of ^11^C compared with ^18^F [165,167]. Indeed, [^18^F]PM-PBB3 (Figure 6c) has greater metabolic stability, higher PET scan throughput, and broader availability than [^11^C]PBB3 (Figure 6a) [169]. Autoradiography studies of frontal AD brain homogenates showed that [^18^F]PM-PBB3 (Figure 6c) has a K_D_ of 7.6 nM [170], comparable to that of [^11^C]PBB3 (Figure 6a). The B_max_ of 5.7 nmol/g in AD brain homogenates thus predicted a high binding potential (B_max_/K_D_ = 750), and the ligand was without binding to Aβ or MAO enzymes in vitro [170].

##### PET Studies PM-PBB3

A PET investigation of [^18^F]PM-PBB3 (Figure 6c) PET in tauopathy model mice demonstrated significantly increased retention in cortical areas, hippocampus, and striatum [168,170]. There was rapid initial uptake of [^18^F]PM-PBB3 (Figure 6c) in brain of the rTg4510 mice, [^18^F]PBB3 (Figure 6b) with a peak radioactivity 1.4 fold higher than for [^18^F]PBB3 (Figure 6b, 2-((1*E*,3*E*)-4-(2-[^18^F]fluoro-)-6-(methylamino)pyridin-3-yl)buta-1,3-dien-1-yl)benzo[d]thiazol-6-ol); the new tracer also had greater metabolic stability compared with its predecessor [170]. Initial conference reports of [^18^F]PM-PBB3 (Figure 6c) PET in humans showed a greater signal-to-background ratio in parametric images compared with [^11^C]PBB3 (Figure 6a) [171]. As opposed to [^11^C]PBB3 (Figure 6a), there was no sign of off-target binding in basal ganglia and thalamus; while there was no sign of binding to MAO, [^18^F]PM-PBB3 (Figure 6c) had more distinct binding in the choroid plexus. On the other hand, [^18^F]PM-PBB3 (Figure 6c) showed high cortical retention in AD patients, matching the regions with hypometabolism to [^18^F]FDG-PET, and a significant positive correlation between cognitive dysfunction (MMSE score) and tracer retention and in all cortical regions except for occipital lobe [79]. Several studies concur in showing that [^18^F]PM-PBB3 (Figure 6c) PET differentiates AD cases from healthy controls with Cohen’s d in the range of two–three in various culprit brain regions (Table 1).

### 2.6. Pyrrolo-Pyridineisoquinolineamines

#### 2.6.1. [^18^F]MK-6240

A screening process yielded 6-[^18^F]fluoro-3-(1*H*-pyrrolo[2,3-c]pyridin-1-yl) isoquinolin-5-amine (Figure 7a, [^18^F]MK-6240) [172]. The tracer showed low binding to MAO-B in molecular docking studies [58,70]. Integrated molecular modeling studies showed that MK6240, similar to T807, has specific binding to site 1 of the tau fibrils, with a preference for core sites over surface sites. [^3^H]MK-6240 showed high-affinity binding (K_D_ = 0.4 nM) to NFT aggregates, with little sign of off-target binding [172]. The B_max_/K_D_ ratio was twice that of [^3^H]AV-1451, predicting more sensitive detection of NFTs. AD brain sections showed [^3^H]MK6240 binding in gray matter regions comparable with the NFT staining pattern, which was displaceable by T808. [^3^H]MK-6240 had no specific binding to subcortical regions of human AD brain slices and did not bind to the entorhinal cortex and hippocampus of the non-AD brain [173]. Saturation binding assays with [^3^H]MK-6240 showed a K_D_ of 0.32 nM and B_max_ of 59 pmol/g in the temporal cortex from AD patients, versus a K_D_ of 0.15 nM and B_max_ of 155 pmol/g in the parietal cortex [174]. Autoradiography of entorhinal and temporal brain sections of AD patients showed strong off-target binding of [^18^F]MK-6240 (Figure 7a) to melanin and neuromelanin-containing cells [175].

[^18^F]MK-6240 (Figure 7a) PET in rhesus monkeys showed favorable brain kinetics and rapid attainment of a homogeneous brain distribution without labeling of the white matter [172,173]. Microdoses of the tracer proved to be safe in human PET studies [176]. Initial [^18^F]MK-6240 (Figure 7a) PET studies showed DVR values > 4 in the AD cortex, which indicates a high affinity to NFTs in vivo accompanied by a low non-displaceable signal [177]. DVR and SUVr values were stable at 90 min after injection, and without off-target binding to the basal ganglia, non-target cortical regions, and choroid plexus. However, the tracer showed off-target binding in the substantia nigra, meninges, ethmoid sinus, and clivus bone, all relative to the pons and inferior cerebellum reference regions. The binding pattern of [^18^F]MK-6240 (Figure 7a) in target areas of Aβ+ patients without dementia and AD patients correlated with Braak staging of NFT accumulation [85,177]. The peak SUV of [^18^F]MK-6240 (Figure 7a) was three–five, indicating high uptake, and the SUVr was about one throughout HC brains versus two–four in NFT-rich regions of AD patients at 60–90 min after tracer injection [178]. Across all subjects, the tracer had adequate test–retest variabilities for various endpoints in NFT-rich brain areas [179].

A comparison with [^18^F]flortaucipir PET (Figure 5a) showed that [^18^F]MK-6240 (Figure 7a) had a two-fold higher dynamic range of SUVr values (SUVr_max_ − SUVr_min_) in AD patients across Braak stages, a property that predicts greater sensitivity for early detection and distinguishing small changes in the rate of AD progression [180]. In that study, [^18^F]MK-6240 (Figure 7a) SUVr values also correlated well with MR-based Braak staging. Cohen’s d values for discrimination of AD and HC groups are in the range of two–four across several studies (Table 1).

#### 2.6.2. [^18^F]GTP1

Although [^18^F]T808 (Figure 5b) had comparable pharmacokinetics to [^18^F]AV-1451 (Figure 5a), the latter tracer was selected for further evaluation due to the substantial in vivo defluorination of [^18^F]T808 (Figure 5b) and accumulation of [^18^F]fluoride in the bones of mice [163]. Isotopic substitution of hydrogen by deuterium on the carbon carrying [^18^F] yielded [^18^F]GTP1 (Figure 7b, Genentech Tau Probe 1), an analogue of [^18^F]T808 (Figure 5b) with reduced defluorination in mice, rhesus monkeys, and humans [181,182]. In vitro studies have shown high selectivity and affinity of [^18^F]GTP1 (Figure 7b) for tau aggregates (K_D_ = 11 nM and B_max_/K_D_ = 12−100), while competitive binding assays with GTP1 against [^18^F]AV-1451 (Figure 5a) in tau-positive tissues indicated a K_i_ of 22 nM [181]. Initial conference reports indicated no binding of [^18^F]GTP1 (Figure 7b) to Aβ and MAO-B [181,182].

Cross-sectional [^18^F]GTP1 (Figure 7b) PET studies in AD patients showed elevated cortical binding in the suspect areas of tau pathology [181,183]. [^18^F]GTP1 (Figure 7b) binding in the temporal lobe correlated with memory deficits and AD severity [183], and the tracer could successfully differentiate AD and HC groups [181,184], with Cohen’s d values in the range of one–two (Table 1).

#### 2.6.3. [^11^C]RO6931643, [^18^F]RO6958948, and [^11^C]RO6924963

In the optimization process of T807 and T808, researchers screened 550 derivatives, among which three compounds could displace [^3^H]T808 binding in human AD brain sections with high affinity: RO6958948 (RO-948; IC_50_ 19 nM), RO6924963 (RO-963; IC_50_ 6 nM), and RO6931643 (RO-643; IC_50_ 10 nM) [185] (Table 1, Figure 7). The tritium-labeled tracers did not bind in AD brain sections lacking tau pathology, and the unlabeled compounds failed to displace [^3^H]florbetapir binding to Aβ in AD tissue sections. RO948 showed tau-to-Aβ selectivity exceeding 500. These three tracers did not exhibit specific binding in brain sections from non-AD tauopathy patients. The compounds were amenable to labeling with [^11^C] and [^18^F]. The three tracers showed no affinity for MAO-A or MAO-B, or other off-target binding sites [185,186].

Initial PET studies of all three tracers conducted in baboons demonstrated appropriate tracer kinetics with peak SUV = 1.2–2.0 within minutes after tracer administration and SUV = 0.5 at 60 min after injection, indicating slow washout, especially for RO963 [185]. The first human PET study of Aβ+ AD patients showed peak SUV values of 3.0, 1.5, and 3.5 in the temporal lobe for [^11^C]RO-963 (Figure 7e, 2-(4-[^11^C]methoxy)phenyl) imidazo [1,2-a]pyridine-7-amine), [^11^C]RO-643 (Figure 7c, *N*-[^11^C]methyl-2-(*m*-tolyl)imidazo[1,2-a] pyrimidin-7-amine), and [^18^F]RO-948 (Figure 7d, 2-(6-([^18^F]fluoro-)pyridine-3-yl)-9*H*-pyrrolo [2,3-b:4,5-c’]dipyridine), respectively [89]. As well as having the greatest uptake, [^18^F]RO-948 (Figure 7d) had the fastest washout from the brain. [^11^C]RO-963 (Figure 7e) was not pursued due to its slow washout, indicative of higher nonspecific binding. Overall, [^18^F]RO-948 (Figure 7d) showed better binding characteristics than [^11^C]RO-643 (Figure 7c), with better uptake between regions expected to have higher tau accumulation. Indeed, [^18^F]RO-948 (Figure 7d) was highly discriminative between AD patients and HCs, and did not exhibit in vivo defluorination. Another [^18^F]RO-948 (Figure 7d) PET study conducted on Aβ+ and Aβ− AD patients demonstrated the absence of off-target binding in the basal ganglia, choroid plexus, thalamus, and white matter, but some retention in the substantia nigra, meninges and retina [186]. The calculated test–retest variability of SUVr in AD patients was 4.6%, indicating good reproducibility. [^18^F]RO-948 (Figure 7d) uptake during three hours of PET recordings conformed well to a reversible binding model. Cohen’s d for differentiation of AD from HC groups was in the range of two–three (Table 1).

### 2.7. Napthyridine Derivatives or Janssen Series (JNJ)

#### 2.7.1. [^18^F]JNJ-311

The ground scaffolds of the Janssen tau series are presented in Figure 8a,b. [^18^F]JNJ-64349311 (Figure 8c, [^18^F]JNJ-311, 6-[^18^F]fluoro-*N*-(2-methyl-4-pyridinyl)-1,5-naphthyridin-2-amine) is a 1,5-napthyridine derivative discovered through the mini-HTS analysis of more than 4000 candidate molecules [187]. The basis of this selection was derived from shape similarity to known selective ligands for tau aggregates, which led to the identification of *N*-(6-methylpyridin-2-yl)quinolin-2-amine. Radioflurorination yielded a ligand with high affinity and selectivity for tau (tau K_i_ = 8 nM, versus Aβ K_i_ > 4400) and no binding to MAO (Table 1).

[^18^F]JNJ-311 (Figure 8c) has suitable brain uptake and washout in mice, and its polar radiometabolites were present only in the plasma compartment [188]. In vitro competition assays of AD brain samples showed displaceable binding to tau aggregates. PET studies in monkeys indicated good penetrance and fast washout, and were without signs of defluorination during a 120 min scan. At the time of writing, there are not tests of this ligand in humans.

#### 2.7.2. JNJ-067

[^18^F]JNJ-64326067 ([^18^F]JNJ-067, *N*-(4-[^18^F]fluoro-5-methylpyridin-2-yl) isoquinoline-6-amine, Figure 8d) is a derivative of [^18^F]JNJ-311 (Figure 8c) with high binding affinity to tau (K_i_ = 2.4 nM) and low off-target binding to Aβ (K_i_ = 3.2 µM), MAO enzymes, and other targets assessed in a standard binding profile [189]. An evaluation of [^3^H]JNJ-067 in an AD brain section showed K_D_ = 0.9 nM and B_max_ = 7.5 nmol/g, which was comparable to findings with [^3^H]T808 (K_D_ = 4.9 nM and B_max_ = 6.3 nmol/g).

Rat and monkey PET studies with [^18^F]JNJ-067 (Figure 8d) showed high initial brain uptake and rapid washout from non-target regions, and no evidence of defluorination at 120 min post-injection [189]. Clinical studies of [^18^F]JNJ-067 (Figure 8d) showed tracer retention in areas of suspected pathology in [^11^C]PIB (Figure 1f) positive AD and MCI patients, but no such retention in HCs [190]. Cohen’s d for differentiation of AD and HC groups was in the range of one–two. There was some off-target binding of [^18^F]JNJ-067 (Figure 8d) in the basal ganglia and white matter. The SUV/DVR ratios ranged from 1.2 to 2.18 in the frontal cortex and 1.21 to 3.09 in the temporal cortex of AD patients, which exceeded the range in healthy controls [90]. However, one AD patient in that study showed no specific tracer signal.

### 2.8. [^18^F]PI-2620

Due to its high affinity for tau deposits and lesser affinity for MAO-A compared with pyrido[4,3-b]indole-derived tracers (AV-1451 (Figure 5a), Figure 7), the pyrrolo[2,3-b:4,5-c’]dipyridine core structures of RO-948 served as the scaffold for some fluoropyridine regioisomers, among which PI-2620 was selected for further studies [191]. A molecular docking study [104] reported a weak interaction between [^18^F]PI-2620 (Figure 7f, 2-(2-([^18^F]fluoro-4-pyridinyl)-9*H*-pyrrolo(2,3-b:4,5-c’)dipyridine) and MAO-B in vitro. [^18^F]PI-2620 (Figure 7f) bound to 3R, 4R, PHF, and K18 fibrils with IC_50_ values ranging from 3 to 20 nM. This encouraged clinical evaluation of [^18^F]PI-2620 (Figure 7f) [191]. Autoradiography studies of AD brain homogenates revealed IC_50_ = 1.8 nM for [^18^F]PI-2620 (Figure 7f) [192].

[^18^F]PI-2620 (Figure 7f) PET showed good kinetics in the human brain, with low test–retest variability of the DVR and SUV endpoints [193]. Cohen’s d for the discrimination between AD patient and HC groups was in the range of two–three in two of three PET studies (Table 1). Early conference reports of human [^18^F]PI-2620 (Figure 7f) PET claimed no off-target binding in areas labeled with [^18^F]AV-1451 (Figure 5a), e.g., choroid plexus and basal ganglia [194,195,196,197]. There was high binding of [^18^F]PI-2620 (Figure 7f) in the substantia nigra and globus pallidus in PSP patients compared with HCs, implying binding to 4R tau filaments, but the binding was higher still in the cortex of AD patients; highest tracer retention in Aβ+ AD patients was in the posterior cingulate gyrus and temporoparietal areas.

The initial human studies of [^18^F]PI-2620 (Figure 7f) showed robust brain uptake and fast washout from non-target areas such as basal ganglia and choroid plexus [93,197,198]. A [^18^F]PI-2620 (Figure 7f) PET study in AD patients determined an appropriate scanning interval of 30–75 min for AD patients [199]. Enhanced tracer retention in AD subjects mostly occurred asymmetrically at the posterior cingulate gyrus, precuneus, parietal, and temporal lobes, with significantly higher DVR and SUVr when compared with HCs [93]. There was a significant correlation between cognitive impairment scores and neocortical tracer uptake. However, [^18^F]PI-2620 (Figure 7f) retention was higher in older HCs [200].

A recent human [^18^F]PI-2620 (Figure 7f) PET study of AD showed no off-target binding in the basal ganglia (i.e., MAO-B binding) in contrast to [^18^F]THK-5351 (Figure 3g) and other first-generation tau tracers [123]. Increased binding in the frontotemporoparietal cortex was noted in Aβ+ MCI and AD patients, whereas Aβ− patients showed no significant increase in [^18^F]PI-2620 (Figure 7f) binding in the cortex, as opposed to the increases in [^18^F]THK5351 (Figure 3g) binding in the same areas. However, one patient with Aβ+ AD showing major atrophy in the bilateral temporal cortex had no cortical increase in [^18^F]PI-2620 (Figure 7f) binding despite elevated [^18^F]THK-5351 (Figure 3g) binding. There was evidence for relatively high [^18^F]PI-2620 (Figure 7f) uptake in vermis, which might also be a general property of second-generation tau PET tracers. Overall, clinical PET studies with [^18^F]PI-2620 (Figure 7f) showed relatively moderate sensitivity for the discrimination of AD and HC groups, with Cohen’s d values around two (Table 1).

### 2.9. Radioiodinated Benzoimidazopyridine (BIP) Derivatives

Benzimidazopyridine (BIP) derivatives are emerging tau tracers for PET and SPECT, notably [^125^I]BIP-NMe_2_ (Figure 9a, 7-[^125^I]iodo-*N*,*N*-dimethylbenzo[[4,5]]imidazo[1,2-a]pyridin-3-amine), which is radioiodinated at the 7-position. Modifications of the 3-position in the BIP scaffold yielded [^18^F]IBIPF1 (Figure 9b, *N*-(2-[^18^F]fluoroethyl)-7-iodo-*N*-methylbenzo[4,5]imidazo [1,2-a]pyridin-3-amine) and [^18^F]IBIPF2 (Figure 9c, *N*-(2-[^18^F]fluoroethyl)-7-iodobenzo[4,5]imidazo [1,2-a]pyridin-3-amine) [201]. Since several PET radioligands such as [^18^F]AV-1451 (Figure 5a) and [^18^F]GTP1 (Figure 7b) also contain the 6,5,6-tricyclic scaffold found in [^125^I]BIP-NMe_2_ (Figure 9a), there is considerable basis for further radioiodinated modifications [202].

Introducing halogens, methoxy, or methyl groups to the 7-position of BIP scaffold resulted in various compounds, including [^18^F]Br-BIPF (Figure 9d, 7-bromo-*N*-(2-[-^18^F]fluoroethyl)-*N*-methylbenzo[[4,5]]imidazo[1,2-a]pyridin-3-amine), [^18^F]Cl-BIPF (Figure 9e, 7-chloro-*N*-(2-[^18^F]fluoroethyl)-*N*-methylbenzo[[4,5]]imidazo[1,2-a]pyridin-3-amine), [^18^F]Me-BIPF (Figure 9f, *N*-(2-[^18^F]fluoroethyl)-*N*,7-dimethylbenzo[[4,5]]imidazo[1,2-a]pyridin-3-amine), and [^18^F]OMe-BIP (Figure 9g, *N*-(2-[^18^F]fluoroethyl)-7-methoxy-*N*-methylbenzo[[4,5]]imidazo[1,2-a]pyridin-3-amine) [203].

Autoradiography studies have indicated the higher effect of modifications in the 3-position of the BIP scaffold on affinity to tau aggregates than modifications of the 7-position [202]. However, [^18^F]Br-BIPF (Figure 9d), [^18^F]Cl-BIPF (Figure 9e), [^18^F]Me-BIPF (Figure 9f), and [^18^F]OMe-BIP (Figure 9g) have also shown appropriate binding to tau aggregates in AD brain sections [201,202]. [^125^I]BIP-NMe_2_ (Figure 9a, [^125^I]7-iodo-3-dimethylaminopyrido[1,2-α]benzimidazole) had higher selectivity than other BIP derivatives, with a Tau/Aβ ratio of 33 [82]. The binding ratio of [^125^I]BIP-NMe_2_ (Figure 9a) in the gray matter of the temporal lobe compared with the frontal lobe was 12.9, and autoradiographic studies in AD brains show selectivity of [^18^F]IBIPF1 (Figure 9b) and [^18^F]IBIPF2 (Figure 9c) tau for aggregates compared with Aβ [204]. [^18^F]IBIPF1 (Figure 9b) binding was absent in normal human temporal lobe sections.

[^125^I]BIP-NMe_2_ (Figure 9a) has shown favorable brain kinetics in healthy mice [205]. [^18^F]IBIPF1 (Figure 9b) showed high initial uptake in mouse brains (6.2% ID/g at two min post-injection) and fast brain washout [206]. Biodistribution studies of [^125^I]21 and [^125^I]22 in normal mice likewise showed favorable pharmacokinetics, potentially superior to that of ([^125^I]BIP-NMe_2_, Figure 9a) [203]. Overall, this study represented two new scaffolds for developing further novel tau tracers, which remain to be studied in AD.

## 3. Discussion and Conclusions

The first successful tau tracer [^18^F]FDDNP (Figure 1a) was an “accidental tau tracer”, having been first intended for detecting Aβ [48]; [^18^F]FDDNP (Figure 1a) has roughly similar affinity for the two molecular targets [52,53] and thus represents the composite of two neurochemical pathologies in AD, despite its possibly lower sensitivity for Aβ [56]. Such discrepancies could reflect factors such as the type of Aβ or tau preparations used. While [^18^F]FDDNP (Figure 1a) is still used for research purposes [207], more selective molecules, notably TAUVID ([^18^F]flortaucipir, Figure 5a), are in clinical use for the diagnosis of AD. As we described above, neither is TAUVID entirely selective for tau aggregates, having a considerable affinity for MAO [125,147]. However, recent studies called into question the contributions of MAO binding to the [^18^F]flortaucipir (Figure 5a) signals from cortical regions [208]. Other tau tracers intended for human PET studies bind to additional targets such as neuromelanin in nigrostriatal dopamine cells, i.e., [^18^F]THK-5351 (Figure 3g), [^18^F]flortaucipir (Figure 5a), and [^18^F]MK-Figure (Figure 7a) [117,141,143,175]. Finally, the degree of lipophilicity of different tracers may contribute to white matter binding and signal-to-noise ratio [63].

To compare the fitness of different tau tracers to discriminate AD and HC groups, we focus on clinical PET studies reporting SUV or SUVr results, with a mean and standard deviation of the endpoint across one or more brain regions, thereby enabling us to calculate Cohen’s d. This may enable an objective judgment of the fitness of the various tau ligands for the specific task of identifying AD at an early disease stage, irrespective of their absolute pharmacological specificity for tau aggregates. Thus, [^18^F]FDDNP (Figure 1a) yielded a Cohen’s d of about 2.3 in the frontal cortex, with lesser values in other cortical regions in mild AD, but a Cohen’s d of around 3 in a more severely affected patient group (Table 1). These large effect sizes indicate considerable fitness for correctly discriminating between the AD and the HC brain. We suppose that its relatively high retention in the frontal cortex reflects the confluence of amyloid and taupathologies. While not imparting definite knowledge about its biochemical nature, the tracer still serves admirably for identifying “net AD pathology”.

Of the various classes of compounds discussed herein, only a few outperformed [^18^F]FDDNP (Figure 1a) for discrimination between AD and HC groups. For example, [^18^F]THK5105 (Figure 3b) had a Cohen’s d as high as 3.6 in the inferior temporal lobe [63], whereas [^18^F]THK5117 (Figure 3d) had a Cohen’s d of 3 in the neocortex and Cohen’s d of 2.7 in the inferior temporal lobe [64]. In other words, these tracers can discriminate the two populations with almost perfect fidelity. However, the THK series has cross-reactivity with MAO [125,126,127,128]. Since astrocytes express MAO-B [209], and since AD and other neurodegenerative diseases are associated with astrocytosis [204], the THK signal may well depict the composite of tau and MAO, again raising the prospect of “net AD pathology”, in analogy to [^18^F]FDDNP (Figure 1a) depicting the composite of tau and Aβ. After treatment of AD patients with MAO-B blockers, PET studies with [^18^F]THK5351 (Figure 3g) showed 50% lower specific binding, which was later confirmed by tissue autoradiography [126,210]. This presents another instance of a fortuitous co-distribution of two targets, namely tau and MAO-B in AD patients, reflecting distinct aspects of AD pathology. 

We conclude with some notes on the limitations of the present literature, and the implicit requirements for improved study design. Thus, the diagnosis of AD in most of the cited studies arose from clinical examination and questionnaires; unsurprisingly, there are very few reports with confirmation of PET results by post mortem histopathological examination, i.e., [128]. We note the generally low group size in most of the cited PET studies, which ranges from about 10 to 50; such group sizes may not capture disease heterogeneity or subtle disease progression, and may inadvertently have resulted in selection bias, with preferential recruitment of AD patients with more advanced disease. Finally, this is a narrative review, which by no means meets the criteria of a systematic review. Thus, we do not include all of the relevant literature, but rather a selection of the papers that enable comparison of Cohen’s d for discriminating AD and HC groups.

## Figures and Tables

**Figure 1 biomolecules-13-00290-f001:**
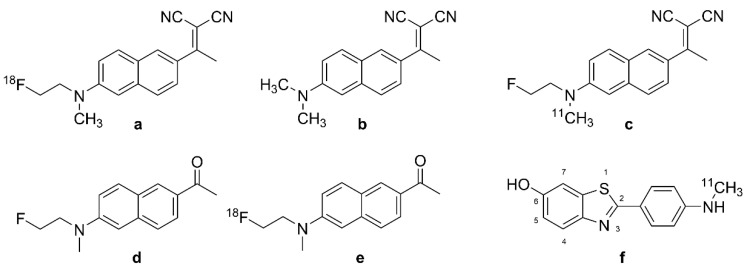
Chemical structure of DDNP analogues and [*N*-methyl-^11^C]-6-OH-BTA-1. (**a**): [^18^F]FDDNP, (**b**): DDNP, (**c**): [^11^C]FDDNP, (**d**): FENE, (**e**): [^18^F]FENE, (**f**): [^11^C]PIB.

**Figure 2 biomolecules-13-00290-f002:**
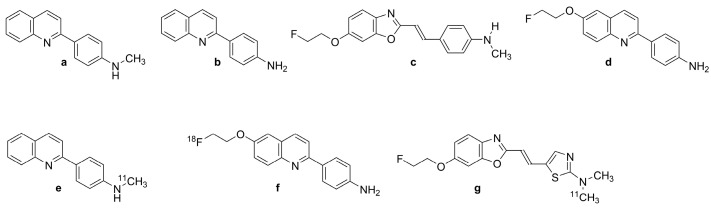
Chemical structure of tau ligands and their labeled derivatives with quinoline and benzoxazole scaffolds. (**a**): BF-158, (**b**): BF-170, (**c**): BF-168, (**d**): BF-242 or THK-523, (**e**): [^11^C]BF-158, (**f**): [^18^F]BF-242 or [^18^F]THK-523, (**g**): [^11^C]BF-227.

**Figure 3 biomolecules-13-00290-f003:**
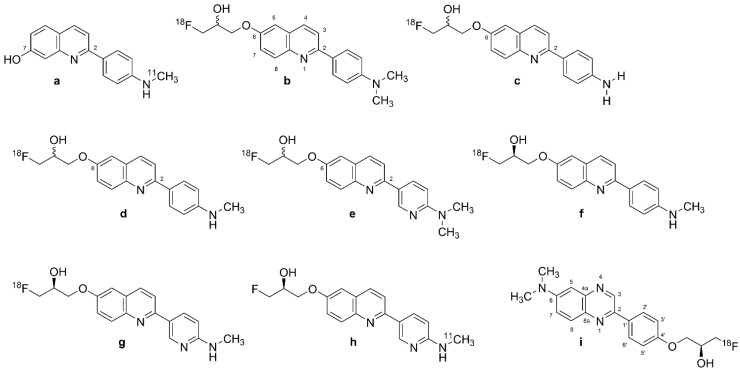
Structure of Tohoku-compounds (labeled 2-arylquinolines, [^18^F]-2-AQ and [^11^C]-2-AQ derivatives). (**a**): [^11^C]THK-951, (**b**): [^18^F]THK-5105, (**c**): [^18^F]THK-5116, (**d**): [^18^F]THK-5117, (**e**): [^18^F]THK-5129, (**f**): [^18^F]THK-5317, (**g**): [^18^F]THK-5351 or [^18^F]GE-216, (**h**): [^11^C]THK-5351, (**i**): [^18^F]-S-16.

**Figure 4 biomolecules-13-00290-f004:**
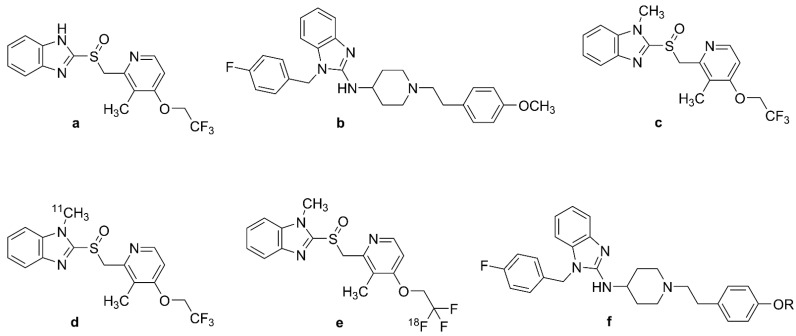
Chemical structures of lansoprazole, astemizole, and their labeled derivatives. (**a**): lansoprazole, (**b**): astemizole, (**c**): *N*-methyl-lansoprazole (NML), (**d**): [^11^C]-*N*-methyl-lansoprazole ([^11^C]NML), (**e**): [^18^F]-*N*-methyl-lansoprazole ([^18^F]NML), (**f**): astemizole-based radiotracers, R = ^11^CH_3_, ^18^FCH_2_CH_2_.

**Figure 5 biomolecules-13-00290-f005:**
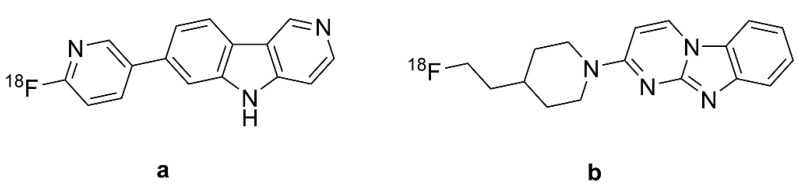
Chemical structures of [^18^F]flortaucipir and [^18^F]T808. (**a**): [^18^F]flortaucipir or [^18^F]T807 or AV-1451, (**b**): [^18^F]T808 or AV-680.

**Figure 6 biomolecules-13-00290-f006:**
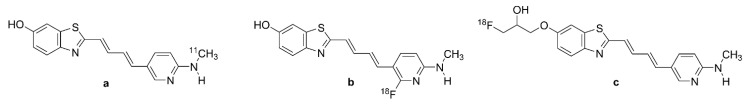
Chemical structure of labeled pyridinyl-butadienyl-benzothiazole derivatives. (**a**): [^11^C]PBB3, (**b**): [^18^F]PBB3, (**c**): [^18^F]PM-PBB3 or [^18^F]APN-1607.

**Figure 7 biomolecules-13-00290-f007:**
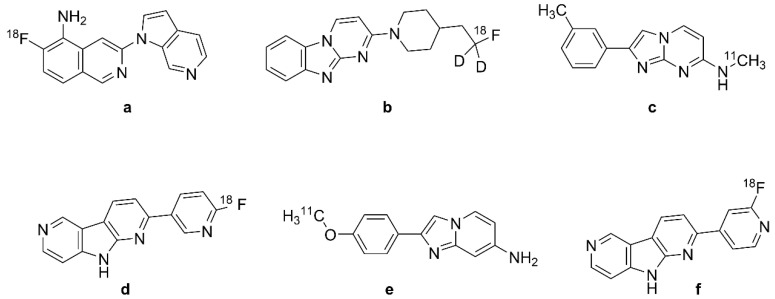
Chemical structures of selected second-generation tau PET radiotracers. (**a**): [^18^F]MK-6240, (**b**): [^18^F]GTP1, (**c**): [^11^C]RO-643 or [^11^C]RO-6931643, (**d**): [^18^F]RO-948 or [^18^F]RO-6958948, (**e**): [^11^C]RO-963 or [^11^C]RO-6924963, (**f**): [^18^F]PI-2620.

**Figure 8 biomolecules-13-00290-f008:**
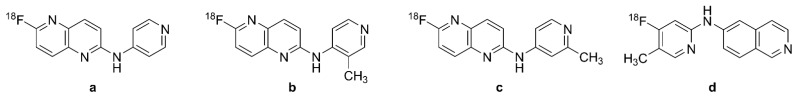
Chemical structures of selected tau PET radiotracers of the Janssen (JNJ) family. (**a**,**b**): 6-[^18^F]fluoro-1,5-naphtyridine derivatives, (**c**): [^18^F]JNJ-311 or [^18^F]JNJ-64349311, (**d**): [^18^F]JNJ-067 or [^18^F]JNJ-64326067.

**Figure 9 biomolecules-13-00290-f009:**
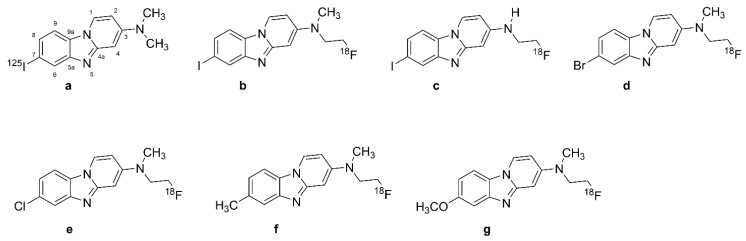
Chemical structures of benzimidazopyridine (BIP) tau radiotracers. (**a**): [^125^I]BIP-NMe_2_, (**b**): [^18^F]IBIPF1, (**c**): [^18^F]IBIPF2, (**d**): [^18^F]Br-BIPF, (**e**): [^18^F]Cl-BIPF, (**f**): [^18^F]Me-BIPF, (**g**): [^18^F]OMe-BIPF.

**Table 1 biomolecules-13-00290-t001:** Comparison between discrimination abilities of tau PET tracers between AD cases and controls in up to three brain regions with highest Cohen’s d values, including as defined by Braak staging system. HC = cognitively healthy elderly controls; AD = Alzheimer’s disease patients; LOAD = late-onset AD; MCI = mild cognitive impairment; MAD = moderate AD; MMSE = mini-mental state examination; SUVr = standard uptake value scaled to the reference tissue; SUVP = standard uptake value scaling with peak alignment; EBM = event-based modeling of sequence of AD pathology.

Tracers	Author (Reference)	Reference Region	n	MMSE ± SD		SUVr of HC	SUVr of AD	Cohen’s d	SUVr of HC		SUVr of AD	Cohen’s d	SUVr of HC	SUVr of AD	Cohen’s d
				HC	AD										
[^18^F]FDDNP	Tauber et al., 2013 [60]	Cerebellum	HC = 8 AD = 7	29.0 ± 1.2	22.1 ± 2.5	Frontal cortex			Occipital cortex				Medial temporal cortex		
						1.06 ± 0.07	1.20 ± 0.05	2.30	0.91 ± 0.07		0.93 ± 0.05	0.32	1.30 ± 0.10	1.32 ± 0.09	0.21
	Tolboom et al., 2009 [61]	Cerebellar gray matter	HC = 13 AD = 14	29 ± 1	23 ± 3	Frontal cortex			Temporal cortex				Parietal cortex		
						0.05 ± 0.04	0.10 ± 0.02	1.58	0.07 ± 0.03		0.10 ± 0.03	1.0	0.03 ± 0.04	0.06 ± 0.03	0.84
	Shin et al., 2008 [62]	Cerebellum	HC = 10 AD = 10	28.4 ± 0.9	13 ± 5	Lateral temporal cortex			Neocortex				Posterior cingulate		
						1.08 ± 0.04	1.24 ± 0.05	3.53	1.06 ± 0.04		1.19 ± 0.05	2.87	1.05 ± 0.05	1.20 ± 0.06	2.71
[^18^F]THK523	Villemagne et al., 2014 [47]	Cerebellar cortex	HC = 10 AD = 10	29.3 ± 1.1	16.7 ± 6.6	Neocortex			Inferior temporal cortex				Insula		
						0.82 ± 0.10	1.13 ± 0.07	3.42	0.96 ± 0.16		1.81 ± 0.58	2.00	0.85 ± 0.16	1.09 ± 0.22	1.30
[^18^F]THK5105	Okamura et al., 2014 [63]	Cerebellar cortex	HC = 8 AD = 8	28.8 ± 1.5	17.3 ± 6.6	Inferior temporal cortex			Superior temporal cortex				Neocortex		
						1.09 ± 0.04	1.32 ± 0.08	3.58	1.04 ± 0.06		1.22 ± 0.07	2.75	1.05 ± 0.05	1.23 ± 0.08	2.68
[^18^F]THK5117	Harada et al., 2015 [64]	Cerebellar cortex	HC = 5 AD = 5	28.7 ± 1.6	18.5 ± 4.6	Neocortex			Posterior cingulate				Inferior temporal cortex		
						1.13 ± 0.05	1.42 ± 0.13	3.05	1.11 ± 0.07		1.43 ± 0.14	2.77	1.15 ± 0.02	1.61 ± 0.23	2.74
[^18^F]THK5317	Fu et al., 2020 [65]	Cerebellum	HC = 6 AD = 5	28.8 ± 0.7	17.2 ± 6.0	Occipital cortex			Lateral temporal cortex				Parietal cortex		
						1.12 ± 0.04	1.36 ± 0.10	3.15	1.15 ± 0.02		1.45 ± 0.14	3	1.07 ± 0.07	1.27 ± 0.09	2.48
[^18^F]THK5351	Chanisa et al., 2021 [66]	Cerebellum	Age ≤ 60 HC = 13 AD = 6	N/A	N/A	Inferior temporal cortex			Occipital cortex				Posterior cingulate		
						1.37 ± 0.04	1.60 ± 0.21	1.52	1.16 ± 0.05		1.35 ± 0.17	1.43	1.45 ± 0.06	1.68 ± 0.24	1.31
			Age > 60 HC = 11 AD = 9	N/A	N/A	Occipital cortex			Precuneus				Inferior temporal cortex		
						1.18 ± 0.12	1.42 ± 0.15	1.76	1.26 ± 0.11		1.47 ± 0.18	1.40	1.48 ± 0.21	1.86 ± 0.32	1.37
	Ezura et al., 2021 [67]	Cerebellar cortex	HC = 9 AD = 10	28.8 ± 1.5	18.9 ± 4.6	Inferior temporal gyrus			Fusiform gyrus				Parahippocampus		
						1.53 ± 0.12	2.11 ± 0.25	3.00	1.60 ± 0.11		2.06 ± 0.19	2.96	2.01 ± 0.12	2.43 ± 0.20	2.54
	Chen et al., 2018 [68]	SUVP	HC = 9 AD = 9	29	20	Temporal cortex			Occipital cortex				Parietal cortex		
						1.57 ± 0.21	1.88 ± 0.22	1.44	1.22 ± 0.13		1.42 ± 0.16	1.37	1.30 ± 0.24	1.46 ± 0.12	0.84
	Kang et al., 2017 [69]	Cerebellar gray matter	HC = 43 AD = 51	28.5 ± 1.6	13.8 ± 6.0	Frontal cortex			Medial temporal cortex				Hippocampus		
						1.35 ± 0.22	2.05 ± 0.34	2.44	2.42 ± 0.27		3.52 ± 0.59	2.39	2.44 ± 0.27	3.40 ± 0.55	2.21
[^18^F]Flortaucipir	Leuzy et al., 2021 [70]	Inferior cerebellar cortex	HC = 638 AD = 159	28.9 ± 1.2	20.4 ± 5.0	Entorhinal cortex			Early tau ROIs				Temporal meta-ROI		
						1.14 ± 0.12	1.74 ± 0.32	2.48	1.19 ± 0.11		1.88 ± 0.46	2.06	1.18 ± 0.11	1.88 ± 0.47	2.05
	Li et al., 2021 [71]	Cerebellar gray matter	HC = 10 AD = 4	29.2 ± 1.3	8.5 ± 6.6	Frontal regions			Lateral temporal region				Parietal region		
						1.32 ± 0.13	2.26 ± 0.12	7.51	1.35 ± 0.14		2.22 ± 0.29	3.82	1.38 ± 0.17	2.59 ± 0.43	3.70
	Wolters et al., 2020 [72]	Cerebellar gray matter	HC = 25 AD/MCI = 53	28 ± 1	23 ± 4	Entorhinal cortex			Limbic region				Neocortex		
						1.1 ± 0.2	1.5 ± 0.2	2.00	1.2 ± 0.1		1.5 ± 0.2	1.89	1.0 ± 0.1	1.4 ± 0.3	1.87
	Chen et al., 2018 [68]	SUVP	HC = 20 AD = 12	29	21	Temporal cortex			Occipital cortex				Frontal cortex		
						1.18 ± 0.16	1.41 ± 0.35	0.84	1.09 ± 0.09		1.17 ± 0.13	0.71	1.06 ± 0.07	1.18 ± 0.23	0.70
	Ossenkoppele et al., 2018 [73]	Cerebellar gray matter	HC = 254 AD = 179	23.6 ± 6.0	20.2 ± 5.5	Entorhinal cortex			Inferior temporal cortex				Temporoparietal cortex		
						1.73 ± 0.31	1.18 ± 0.2	2.10	2.09 ± 0.56		1.23 ± 0.21	2.03	1.89 ± 0.53	1.15 ± 0.18	1.86
	Pontecorvo et al., 2017 [74]	Cerebellar gray matter	Aβ + OC = 58 Aβ + AD = 48	29.5 ± 0.5	22.1 ± 3.7	Fusiform gyrus			Anterior parahippocampal gyrus				Temporal cortex		
						1.15 ± 0.10	1.66 ± 0.30	2.28	1.07 ± 0.13		1.49 ± 0.24	2.17	1.11 ± 0.09	1.64 ± 0.40	1.82
	Cho et al., 2016 [75]	Cerebellar cortex	HC = 20 AD = 20	27.5 ± 2.1	16.9 ± 6.6	Entorhinal cortex			Parahippocampal cortex				Inferior temporal cortex		
						1.22 ± 0.16	1.80 ± 0.33	2.26	1.20 ± 0.17		1.49 ± 0.25	1.99	1.71 ± 0.34	1.22 ± 0.19	1.79
[^11^C]PBB3	Kitamura et al., 2018 [76]	Cerebellar gray matter	HC = 9 AD = 17	29.4 ± 0.7	21.4 ± 6.5	Occipital cortex			Parietal cortex				Lateral temporal cortex		
						0.90 ± 0.03	1.10 ± 0.1	2.70	0.83 ± 0.05		1.03 ± 0.1	2.52	0.95 ± 0.04	1.12 ± 0.11	2.05
	Shimada et al., 2017 [77]	Cerebellar cortex	HC = 18 AD = 17	28.9 ± 1.2	16.1 ± 5.1	Mean cortical									
						0.91 ± 0.06	1.10 ± 0.07	2.91							
[^18^F]PM-PBB3	Hsu et al., 2020 [78]	Cerebellar gray matter	HC = 12 AD = 10	29.3 ± 0.9	12.5 ± 8.9	Parahippocampus			Temporal region				Posterior cingulate gyrus		
						0.97 ± 0.11	2.06 ± 0.69	2.20	0.99 ± 0.10		2.46 ± 0.95	2.17	1.02 ± 0.09	2.63 ± 1.09	2.08
	Lu et al., 2020 [79]	Cerebellar cortex	HC = 11 AD = 19	N/A	17.0 ± 7.6	Temporal lobe			Frontal lobe				Occipital lobe		
						0.91 ± 0.06	1.64 ± 0.50	2.06	0.85 ± 0.06		1.43 ± 0.42	1.92	0.95 ± 0.06	1.58 ± 0.47	1.91
[^18^F]MK6240	Therriault et al., 2022 [80]	Inferior cerebellar cortex	HC = 179 AD = 65	29.1 ± 1.4	19.7 ± 6.2	Temporal meta-ROIs									
						1.06 ± 0.15	2.82 ± 1.03	2.39							
	Ashton et al., 2021 [81]	Inferior cerebellar cortex	HC = 159 AD = 42	29.1 ± 1.1	18.5 ± 5.7	Braak I–II			Braak III–IV				Braak V-VI		
						0.97 ± 0.2	1.98 ± 0.6	2.25	0.95 ± 0.1		2.61 ± 1.1	2.12	0.97 ± 0.1	2.25 ± 2.1	0.86
	Leuzy et al., 2021 [70]	Inferior cerebellar cortex	HC = 218 AD = 50	29.2 ± 1.0	18.4 ± 5.9	Early tau ROIs			Entorhinal cortex				Temporal meta-ROIs		
						0.87 ± 0.12	2.80 ± 0.64	4.19	0.93 ± 0.23		2.4 ± 0.56	3.48	0.86 ± 0.11	2.84 ± 0.66	3.42
	Therriault et al., 2021 [82]	Inferior cerebellar cortex	HC = 131 AD = 25	29.1 ± 1.2	20.1 ± 5.7	Braak III–IV			Braak V–VI				Braak I–II		
						1.04 ± 0.17	2.89 ± 0.87	2.95	1.09 ± 0.13		2.47 ± 0.96	2.01	0.99 ± 0.29	2.02 ± 0.86	1.60
	Therriault et al., 2021 [83]	Inferior cerebellar cortex	HC = 166 AD = 62	29.1 ± 1.0	19.2 ± 6.2	AD signature meta-ROI			Braak I–II						
						1.08 ± 0.24	3.3 ± 1.4	2.21	0.98 ± 0.22		2.04 ± 0.75	1.91			
	Tissot et al., 2021 [84]	Inferior cerebellar cortex	HC = 143 AD = 26	29.0 ± 1.2	19.3 ± 7.1	Braak III–IV			Braak I–II				Braak V-VI		
						1.07 ± 0.12	2.51 ± 0.91	2.21	0.97 ± 0.18		1.63 ± 0.40	2.12	1.10 ± 0.14	2.24 ± 0.87	1.82
	Pascoal et al., 2020 [85]	Inferior cerebellum	HC = 101 LOAD = 21	29.1 ± 1.1	21.2 ± 5.3	Braak II			Braak I				Braak III		
						1.14 ± 0.15	2.6 ± 0.97	2.10	1.13 ± 0.17		3.05 ± 1.32	2.04	1.17 ± 0.1	2.6 ± 1.18	1.93
[^18^F]GTP1	Barthelemy et al., 2022 [86]	Cerebellar gray matter	Aβ+ HC = 5 MAD = 10	29.0 ± 0.7	17.6 ± 2.7	Temporal meta-ROIs			Braak III–IV				Braak I–II		
						1.26 ± 0.05	1.67 ± 0.32	1.79	1.2 ± 0.05		1.55 ± 0.35	1.4	1.27 ± 0.04	1.48 ± 0.26	1.12
[^18^F]RO948	Leuzy et al., 2022 [87]	Inferior cerebellar cortex	Aβ− HC = 137 AD = 63	28.9 ± 1.2	19.7 ± 4.2	EBM stage I			EBM stage II				EBM stage III		
						0.97 ± 0.13	1.71 ± 0.36	2.80	1.27 ± 0.11		2.62 ± 0.94	2.01	1.26 ± 0.12	2.37 ± 0.91	1.71
	Leuzy et al., 2021 [70]	Inferior cerebellar cortex	HC = 208 AD = 142	28.7 ± 1.2	20.3 ± 4.1	Entorhinal cortex				Early tau ROIs			Temporal meta-ROIs		
						1.23 ± 0.22	2.00 ± 0.40	3.78	1.23 ± 0.18		2.15 ± 0.66	1.90	1.22 ± 0.18	2.13 ± 0.66	1.88
	Leuzy et al., 2020 [88]	Inferior cerebellar cortex	HC = 257 AD = 100	29 ± 1.1	20 ± 4.3	Braak stage V–VI			Entorhinal cortex				Braak stage I-IV		
						1.06 ± 0.10	1.53 ± 0.21	2.85	1.16 ± 0.22		l2.02 ± 0.40	2.66	1.17 ± 0.15	2.19 ± 0.58	2.40
	Wong et al., 2018 [89]	Cerebellar cortex	HC = 5 AD = 11	N/A	20.8 ± 2.8	Parahippocampus			Entorhinal area				Anterior temporal lobe		
						0.93 ± 0.2	1.95 ± 0.7	1.98	0.93 ± 0.2		1.91 ± 0.7	1.89	1.01 ± 0.1	1.71 ± 0.6	1.62
[^18^F]JNJ067	Schmidt et al., 2020 [90]	Ventral cerebellar cortex	HC = 5 AD = 5	28.4 ± 0.5	19.2 ± 5.9	Medial temporal cortex			Amygdala				Occipital cortex		
						1.18 ± 0.09	1.76 ± 0.47	1.71	0.96 ± 0.12		1.51 ± 0.46	1.63	1.14 ± 0.10	1.54 ± 0.40	1.37
[^18^F]PI2620	Bun et al., 2022 [91]	Cerebellum	HC = 7 AD = 7	28.4 ± 1.5	18.7 ± 5.1	Right amygdala			Left amygdala				Left hippocampus		
						1.14 ± 0.08	1.98 ± 0.31	3.63	1.19 ± 0.14		2.07 ± 0.44	2.66	1.42 ± 0.14	1.81 ± 0.21	2.08
	Jantarato et al., 2021 [92]	Cerebellum	HC = 26 AD = 7	27.3	18.4	Occipital cortex			Precuneus				Caudate		
						1.1 ± 0.12	1.14 ± 0.21	0.93	1.02 ± 0.12		1.27 ± 0.45	0.75	0.80 ± 0.13	0.7 ± 0.14	0.74
	Mueller et al., 2020 [93]	Cerebellum	HC = 10 AD = 12	29	20.5	Inferior temporal			Fusiform gyrus				Occipital		
						1.08 ± 0.08	1.70 ± 0.32	2.59	1.08 ± 0.09		1.57 ± 0.24	2.55	1.06 ± 0.07	1.39 ± 0.19	2.13
[^18^F]S16	Fu et al., 2022 [65,94]	Cerebellar gray matter	HC = 6 AD = 5	28.8 ± 0.7	17.2 ± 6.0	Occipital cortex			Lateral temporal cortex				Medial temporal cortex		
						1.10 ± 0.07	1.21 ± 0.08	1.46	1.22 ± 0.09		1.30 ± 0.10	0.84	1.11 ± 0.10	1.18 ± 0.12	0.63
	Wang et al., 2021 [95]	Cerebellum	HC = 6 AD = 15	29.8 ± 0.3	18.6 ± 5.3	Parietal cortex			Posterior cingulate and precuneus				Occipital cortex		
						1.10 ± 0.05	1.35 ± 0.19	1.78	1.17 ± 0.04		1.35 ± 0.14	1.70	1.05 ± 0.03	1.19 ± 0.12	1.49

## Data Availability

Not applicable.

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
