# Peer review of "The Sensitivity of Tau Tracers for the Discrimination of Alzheimer’s Disease Patients and Healthy Controls by PET"

_biomolecules, 2023, doi:10.3390/biom13020290_

Round 1

Reviewer 1 Report

  The present review is well-written and clear, the argument is interesting and in line with the scopus of the journal.

Major suggestions:
In the main text there are several references to Table 1 but there isn't a table in the maniscript. Please, revise it.

A table/paragraph that report Cohen's d values described in the text is required. Please, add a paragraph or a table in the manuscript that reports Cohen’s d for the contrast between tracer binding in groups of healthy controls HCs and AD patients.

Minor suggestions:
A table that summarize the Tau tracers described, with a brief overview of the study settings (preclinical/clinical/multi-centric trials etc. and the sample's size in the clinic trials) and main findings would benefit the manuscript.

I suggest to reformat the figures. I suggest to nominate the sub-figures in letters. For example, for the Figure 1, I suggest to change the numbers in letters (a,b,c,d,e,f) and include the description of sub-figures in figure's caption. Moreover, I suggest to cite subfigures in the main text in this format: Fig 1a.
Moreover, I suggest to avoid to cite the figure in the title of the section/paragraph/subparagraph but only in the text.  

Author Response

Major suggestions: In the main text there are several references to Table 1 but there isn't a table in the manuscript. Please, revise it.
A table/paragraph that report Cohen's d values described in the text is required. Please, add a paragraph or a table in the manuscript that reports Cohen’s d for the contrast between tracer binding in groups of healthy controls HCs and AD patients.

Reply: I am mortified to see that I failed to upload the table in the original submission. The table is now included in the body of the manuscript

Minor suggestions:
A table that summarize the Tau tracers described, with a brief overview of the study settings (preclinical/clinical/multi-centric trials etc. and the sample's size in the clinic trials) and main findings would benefit the manuscript.

I suggest to reformat the figures. I suggest to nominate the sub-figures in letters. For example, for the Figure 1, I suggest to change the numbers in letters (a,b,c,d,e,f) and include the description of sub-figures in figure's caption. Moreover, I suggest to cite subfigures in the main text in this format: Fig 1a.
Moreover, I suggest to avoid to cite the figure in the title of the section/paragraph/subparagraph but only in the text.  

Reply: We have corrected the figures accordingly.

Reviewer 2 Report

The authors in manuscript entitled “The sensitivity of tau tracers for the descrimination of Alzheimer’s disease patients and healthy controls by PET” have explored the Molecular imaging of tau by positron emission tomography (PET) began with the development of [18F]FDDNP, an amyloid β tracer with off-target binding to tau, which obtained re-gional specificity through the differing distributions of amyloid β and tau in AD brain.

There are some minor issue with English of the paper, English must be improved, Introduction part should be elaborative and conclusion part should be more precise,  if these issues are going to resolve then the quality of the paper is suitable for publication.

Author Response

Alzheimer’s disease patients and healthy controls by PET” have explored the Molecular imaging of tau by positron emission tomography (PET) began with the development of [18F]FDDNP, an amyloid β tracer with off-target binding to tau, which obtained re-gional specificity through the differing distributions of amyloid β and tau in AD brain.
There are some minor issue with English of the paper, English must be improved, Introduction part should be elaborative and conclusion part should be more precise,  if these issues are going to resolve then the quality of the paper is suitable for publication.

Reply: By moving several sections of text, we have expanded the Introduction from 300 words to 700 words. We agree that this leads to a better balance and structure of the main text. Through considerable truncation, the discussion and conclusions section now has 700 words, as compared to 1030 words in the original submission. We have carefully checked for language.
The present review is well-written and clear, the argument is interesting and in line with the scopus of the journal.

Round 2

Reviewer 1 Report

This article is well-written and clear.

All the changes requested have been performed.

It is very interesting and I suggest the publication.